# Bandit Learning in Matching Markets with Switching Cost

## Abstract

We study the bandit learning problem in two-sided matching markets. While existing works successfully derive sub-linear bounds for the player-optimal regret, they typically assume cost-free switching and may incur up to $O(T)$ switches over a time horizon of length $T$. Such frequent reassignments are impractical in real-world applications since switching is usually costly and disruptive. To address this limitation, we explicitly incorporate switching costs into the decision-making process and aim to minimize player-optimal stable regret under a switching-cost budget. We first consider a setting with unit switching cost, where each switch incurs a fixed cost. We propose a cost-aware algorithm that achieves the same regret bound of $O(\log T/\Delta^2)$ as previous approaches while reducing the total number of switches to $O(\log T)$, where $\Delta$ is the players' minimum preference gap. Furthermore, we show that by slightly relaxing the regret to $O(\sqrt{T/\Delta^2})$, the total number of switches can be reduced to $O(\log \log T)$; in the extreme case, with only $O(1)$ switches, the algorithm still guarantees a regret of $O(T^{2/3})$. We also generalize this approach to heterogeneous switching cost setting by leveraging the shortest Hamiltonian path orderings and provide analogous theoretical guarantees.

## 1 Introduction

Two-sided matching markets are foundational in numerous domains, including labor markets, school admissions, and online matching platforms (Roth, 1984; Gale & Shapley, 1962; Abdulkadiroğlu & Sönmez, 1999). The concept of stable matching plays a crucial role in ensuring the long-term stability of such markets and has been extensively examined in the literature (Gale & Shapley, 1962; Roth & Sotomayor, 1992). Traditionally, matching algorithms operate under the assumption that participants' preferences are fully known in advance. However, in many real-world settings, market participants often face uncertainty regarding their own preferences and must learn them through repeated interactions. The multi-armed bandit (MAB) framework offers a well-established model for capturing this learning process over time (Auer et al., 2002; Lattimore & Szepesvári, 2020).

Recent research has investigated the intersection of matching markets and multi-armed bandit (MAB) frameworks (Liu et al., 2020; 2021; Sankararaman et al., 2021; Basu et al., 2021; Kong et al., 2022; Maheshwari et al., 2022; Ghosh et al., 2022; Kong & Li, 2023; Kong et al., 2024; Lin et al., 2024). These studies typically model the two sides of the market as players and arms in the MAB setting, respectively. In this formulation, players learn their uncertain preferences over arms through repeated matching processes. The goal is to identify a stable matching as far as possible, commonly formalized as minimizing stable regret which is defined as the cumulative difference between the satisfaction derived from the stable partner and that from the actual matched partner during the learning process.

A key challenge in this setting arises from conflicts between multiple players vying for the same arms. Most existing approaches have been able to guarantee convergence only to the player-pessimal stable matching, where each player is matched with their least-preferred stable partner (Liu et al., 2021; Kong et al., 2022; Sankararaman et al., 2021; Basu et al., 2021; Maheshwari et al., 2022). Until recently, Zhang et al. (2022); Kong & Li (2023); Kong et al. (2024) have proposed algorithms that achieve player-optimal stability, ensuring more favorable outcomes for participants.

Despite the importance of these results, the corresponding algorithms operate under a crucial simplifying assumption: switching between different arms is cost-free—that is, players incur no penalty when

changing partners. To quickly collect observations over arms and avoid frequent conflicts among players, the state-of-the-art approaches all follow a round-robin scheme (Zhang et al., 2022; Kong & Li, 2023; Kong et al., 2024), where players match different arms in each round. In the worst-case scenario, this strategy can lead to up to $O(T)$ switching operations over a time horizon $T$.

However, in real-world applications, frequent switching between different partners often incurs substantial operational and economic costs. For instance, in server deployment scenarios, repeatedly terminating and launching new tasks can lead to increased latency, resource fragmentation, and performance degradation. Such disruptions can significantly affect system reliability, particularly in latency-sensitive or high-availability environments (Makhlouf, 2020). In online platforms such as freelance marketplaces or delivery services, continuous reassignment of partners may reduce user satisfaction and introduce considerable coordination overhead (Ma et al., 2021; Correa et al., 2022; Fulker & Riedl, 2023; Echenique et al., 2024). Likewise, in online labor markets or crowdsourcing platforms, frequent job transitions can entail relocation expenses, onboarding and training efforts, and potential reputational risks (Yiu et al., 2024; Hajiaghayi et al., 2024; Lechowicz et al., 2024; Miserendino et al., 2025). Consequently, although learning participant preferences through repeated interactions is crucial, minimizing switching costs is a key consideration for the practical deployment of learning-based matching algorithms.

In this paper, we formally incorporate switching costs into the players' decision-making process. Our objective is to minimize the player-optimal stable regret under a constrained switching cost budget. A central challenge arises from the inherent trade-off between learning efficiency, player conflicts, and switching costs: rapid learning typically requires a player to frequently switch between arms to collect information and make timely decisions, yet excessive switching both incurs significant costs and increases the risk of conflicts. To address this challenge, we introduce the principle of consecutive exploration, which ensures a predictable and controllable switching cost, and complement it with a phased monitoring mechanism to determine when exploration is sufficient. Different choices of phase length induce distinct trade-offs between switching cost and regret. By carefully tuning the phase length, we obtain the following results:

- We match the performance of existing methods by guaranteeing $O(K \log T / \Delta^2)$ player-optimal stable regret, while significantly reducing the worst-case switching cost from $O(T)$ to $O(K \log T)$. Here $K$ is the number of arms, $T$ is the horizon, and $\Delta$ is the player's minimum preference gap.

- Furthermore, if the regret requirement is relaxed to $O(\sqrt{KT/\Delta^2})$, the switching cost can be further reduced to $O(K \log \log T)$.

- In the extreme case where the player's switching budget is limited to $O(K)$, we provide an $O(T^{2/3})$ upper bound on the relaxed $\alpha$-fraction regret.

We further extend our analysis to the general switching cost setting, where transition costs vary across players and arm pairs. This extension introduces additional challenges in coordinating players to minimize switching costs while avoiding collisions, a problem that is NP-hard. We formalize this as a multi-player shortest Hamiltonian path ordering problem, which may be of independent interest in domains such as distributed systems, robotics, and network design. For this problem, we develop approximation algorithms with provable performance guarantees. Building on these results, we further show that our framework achieves theoretical guarantees analogous to those obtained in the unit-cost setting. Table 1 summarizes the regret bounds and switching costs achieved by our algorithm in comparison with existing approaches.

## 2 RELATED WORK

Low-switching-cost bandit algorithms have been studied primarily in a single-player setting. The literature can be broadly divided into two directions: one focuses on designing algorithms that explicitly satisfy switching cost constraints (Gao et al., 2019; Esfandiari et al., 2021; Simchi-Levi & Xu, 2023), while the other incorporates switching costs into the per-round regret and jointly optimizes cumulative cost and reward (Dekel et al., 2014; Koren et al., 2017). Our work aligns with the former, aiming to develop algorithms in which switching is subject to hard constraints—such as physical limitations or budgetary restrictions. In the classical single-player stochastic MAB setting, it is known that a regret of $O(\log T / \Delta)$ can be achieved with switching cost $O(\log T)$, and a regret of $O(\sqrt{KT})$ can be achieved with switching cost $O(\log \log T)$ (Gao et al., 2019).

Extending these algorithms to matching markets introduces a fundamental and previously unexplored three-way trade-off among learning efficiency, player conflict, and switching cost. Existing bandit algorithms for matching markets address only the interplay between the first two factors, without examining how switching costs reshape the learning process and the dynamics of player interactions. In what follows, we briefly review the regret guarantees and switching costs achieved in these works. Based on how they address the exploration–exploitation trade-off, existing bandit algorithms can be categorized into three main classes: UCB- (Liu et al., 2020; 2021), TS- (Kong et al., 2022) and ETC-type (Liu et al., 2020; Basu et al., 2021; Zhang et al., 2022; Kong & Li, 2023; Kong et al., 2022).

UCB- and TS-type algorithms select arms based on previously collected feedback and collision information with other players. Prior to convergence, players can be viewed as exploring different arms, implying that the switching cost is of the same order as the regret. In general decentralized matching markets, the regret and the expected switching cost are both of order $O(N^5 K^2 \log^2 T / \epsilon^{N^4} \Delta^2)$ (Liu et al., 2021; Kong et al., 2022). And theoretical results only guarantee the convergence to the player-pessimal stable matching.

Table 1: Comparisons of regret bounds and switching costs with existing works. $*$ represents the player-optimal stable regret and bounds without labeling $*$ are for player-pessimal stable regret, $\#$ represents the $\alpha$-fraction regret (Definition 2). $T$ denotes the time horizon, $N$ and $K$ are the numbers of players and arms, respectively, with $N \leq K$. $\epsilon$ is determined by the algorithm's hyperparameter, and $\Delta$ represents a preference gap. Noting that the definition of $\Delta$ varies slightly in different works. Please see Kong et al. (2024) for a detailed comparison. Most existing algorithms incur a worst-case switching time of $O(T)$. Some ETC-type method (Liu et al., 2020) requires $O(K \log T / \Delta^2)$, while Phased-ETC (Basu et al., 2021) achieves $O(K \log^{1+\epsilon} T)$. ETCO in Lin et al. (2024) attains a worst-case bound of $O((K \log T)^{\frac{1}{3}} T^{\frac{2}{3}})$. By contrast, Algorithm 1 achieves worst-case switching time of $O(K \log \log T), O(K \log T / \Delta^2)$ and $O(K)$ respectively, under different instantiations.

| | **Regret bound** | **Expected switching time** |
|---|---|---|
| Liu et al. (2020) | $O(K \log T / \Delta^2) *$ 
 $O(NK \log T / \Delta^2)$ | $O(K \log T / \Delta^2)$ 
 $O(NK \log T / \Delta^2)$ |
| Liu et al. (2021) | $O\left(\frac{N^5 K^2 \log^2 T}{\epsilon^{N^4} \Delta^2}\right)$ | $O\left(\frac{N^5 K \log^2 T}{\epsilon^{N^4} \Delta^2}\right)$ |
| Sankararaman et al. (2021) | $\Omega(NK \log T / \Delta^2)$ 
 $\Omega(\max\{N \log T / \Delta^2, K \log T / \Delta\})$ | $O(K \log T / \Delta^2)$ |
| Basu et al. (2021) | $O\left(K \log^{1+\epsilon} T + 2^{\left(\frac{1}{\Delta^2}\right)^{\frac{1}{\epsilon}}}\right) *$ | $O(K \log^{1+\epsilon} T)$ |
| Kong et al. (2022) | $O\left(\frac{N^5 K^2 \log^2 T}{\epsilon^{N^4} \Delta^2}\right)$ | $O\left(\frac{N^5 K^2 \log^2 T}{\epsilon^{N^4} \Delta^2}\right)$ |
| Zhang et al. (2022) | $O(K \log T / \Delta^2) *$ | $O(K \log T / \Delta^2)$ |
| Kong & Li (2023) | $O(K \log T / \Delta^2) *$ | $O(K \log T / \Delta^2)$ |
| Lin et al. (2024) | $O((K \log T)^{\frac{1}{3}} T^{\frac{2}{3}}) \#$ | $O((K \log T)^{\frac{1}{3}} T^{\frac{2}{3}})$ |
| Kong et al. (2024) | $O(N^2 \log T / \Delta^2 + K \log T / \Delta)$ | $O(N^2 \log T / \Delta^2 + K \log T / \Delta)$ |
| Ours | $O(\sqrt{KT \log T / \Delta^2}) *$ 
 $O(K \log T / \Delta^2) *$ 
 $O((K \log T)^{\frac{1}{3}} T^{\frac{2}{3}})$ | $O(K \log \log(\frac{\log T}{\Delta^2}))$ 
 $O(K \log(\frac{\log T}{\Delta^2}))$ 
 $O(K)$ |

ETC-type algorithms, by explicitly coordinating players' arm selections, typically guarantee convergence to the player-optimal stable matching. Liu et al. (2020) study the ETC algorithm in the centralized setting. An exploration budget $h$ is specified, and players explore the $K$ arms in a round-robin manner, with each arm selected $h$ times. The switching cost is $O(K \log T / \Delta^2)$, while achieving the same order of regret. Extending this line of work, Basu et al. (2021) consider the decentralized setting and propose a phased ETC algorithm. In each phase, players first explore different arms for a fixed budget and then apply GS to obtain a stable matching. The switching

cost is $O(K \log^{1+\epsilon} T)$, and the regret includes an additional exponential constant term. Subsequent works improve upon this algorithm by eliminating unnecessary exploitation before the arm estimates are sufficiently accurate. These algorithms follow an explore-then-GS framework: players initially explore different arms and continuously test whether the exploration is adequate, after which they run GS to determine a stable matching (Zhang et al., 2022; Kong & Li, 2023; Kong et al., 2024). Most of the switching occurs during the exploration phase. The expected switching cost and regret are both $O(K \log T / \Delta^2)$ (Kong & Li, 2023; Zhang et al., 2022) or $O(N^2 \log T / \Delta^2)$ (Kong et al., 2024), whereas in the worst case—when the stopping condition never triggers—the switching cost can be as large as $O(T)$.

## 3 PRELIMINARIES

Without loss of generality, suppose there are $N$ players and $K$ arms in the market. Let $\mathcal{N} = \{p_1, p_2, \ldots, p_N\}$ denote the player set and $\mathcal{K} = \{a_1, a_2, \ldots, a_K\}$ denote the arm set. To ensure that each player has the opportunity to be matched, we assume that $N \leq K$, as is commonly assumed in existing works (Liu et al., 2020; 2021; Zhang et al., 2022; Kong & Li, 2023; Kong et al., 2024).

The preference of player $p_i$ over arm $a_j$ is modeled as a preference value $\mu_{i,j} \in (0, 1]$. When comparing two arms $a_j$ and $a_{j'}$, the inequality $\mu_{i,j} > \mu_{i,j'}$ indicates that player $p_i$ prefers arm $a_j$ over arm $a_{j'}$. Consistent with existing works (Roth & Sotomayor, 1992; Liu et al., 2020; 2021; Kong & Li, 2023; Zhang et al., 2022; Kong et al., 2024), we assume that players have distinct preferences for different arms, i.e., $\mu_{i,j} \neq \mu_{i,j'}$ whenever $a_j \neq a_{j'}$. As observed in real-world applications such as the Upwork labor market, players' preferences are typically unknown initially but can be learned through repeated interactions with arms. Similarly, each arm $a_j$ has a preference vector $(\pi_{j,i})_{i \in [N]}$ over players, where $\pi_{j,i} > \pi_{j,i'}$ implies that arm $a_j$ favors player $p_i$ over player $p_{i'}$. For simplicity, we assume that arms know their preferences as in existing works (Liu et al., 2020; Zhang et al., 2022; Kong & Li, 2023; Kong et al., 2024; Lin et al., 2024).

Players interact with arms through iterative matching processes. Specifically, in each time slot $t = 1, 2, \ldots$, player $p_i$ selects an arm $A_i(t) \in \mathcal{K}$ to propose. Arms then select the most preferred proposal to accept. Specifically, let $A_j^{-1}(t) = \{p_i : A_i(t) = j\}$ denote the set of players who propose to arm $a_j$. Arm $a_j$ would accept the most preferred player, i.e., $\arg\max_{p_i \in A_j^{-1}(t)} \pi_{j,i}$. If player $p_i$ is accepted, it receives a random reward $X_i(t)$ that characterizes the matching process, which can be treated as 1-subgaussian random variable with mean $\mu_{i,A_i(t)}$. Otherwise, the player receives a reward of $X_i(t) = 0$. For convenience, let $\bar{A}(t) = \{(i, \bar{A}_i(t) : i \in [N]\}$ be the final matching at $t$, i.e., $\bar{A}_i(t) = A_i(t)$ if $p_i$ is ultimately matched with the proposed arm $A_i(t)$, and $\bar{A}_i(t) = \emptyset$ otherwise.

To efficiently collect matching observations across different arms, existing approaches commonly adopt a round-robin exploration strategy (Liu et al., 2020; Basu et al., 2021; Zhang et al., 2022; Kong & Li, 2023; Kong et al., 2024). Such strategies ensure that players can learn as much about unknown preferences as possible. However, frequent switching between different arms can incur high switching costs. For instance, in the context of server resource allocation, the constant deployment and withdrawal of computing tasks not only significantly increase the operational burden but also lead to system instability and even service interruptions. In this work, we explicitly incorporate the costs incurred when players switch between different matching partners into the learning process. For each player $p_i$, we define the switching cost function as $c_i := (c_{i,j,j'})_{j,j' \in \mathcal{K} \cup \emptyset}$. Specifically, when $p_i$ switches its matching partner from $a_j$ to $a_{j'}$, a cost of $c_{i,j,j'} > 0$ is incurred. It is natural to assume that the costs satisfy $c_{i,j,\emptyset} + c_{i,\emptyset,j'} = c_{i,j,j'}$ and $c_{i,\emptyset,\emptyset} = 0$.

Stability is a critical objective in matching markets, as it ensures that once a matching is implemented, no blocking pairs emerge. A matching $\bar{A}(t) = \{(i, \bar{A}_i(t)) : i \in [N]\}$ is considered stable if no player-arm pair $(p_i, a_j)$ exists such that both $p_i$ and $a_j$ would prefer to be matched with each other rather than remaining with their current assignment. Formally, stability requires that there is no blocking pair $(p_i, a_j)$ such that $\mu_{i,j} > \mu_{i,\bar{A}_i(t)}$ and $\pi_{j,i} > \pi_{j,\bar{A}_j^{-1}(t)}$. Since multiple stable matchings can exist, let $M := \{m : m \text{ is a stable matching}\}$ denote the set of all stable matchings. The ideal goal is to achieve the player-optimal stable matching, $m^* = \{(i, m_i^*) : i \in [N]\} \in M$, which ensures that $\mu_{i,m_i^*} \geq \mu_{i,m_i}$ for any player $p_i$ and any stable matching $m \in M$. To assess the convergence performance to the player-optimal stable matching, we define the player-optimal stable regret as the cumulative difference between the reward in the player-optimal stable matching and the rewards

received during interactions:

$$Reg_i(T) = \mathbb{E}\left[\sum_{t=1}^{T}(\mu_{i,m_i^*} - X_i(t))\right],$$

where the expectation is derived from the randomness of player strategies and reward generations.

Given a switching cost budget $S > 0$, our objective is to design an algorithm that minimizes the player-optimal stable regret $Reg_i(T)$ for any player $p_i \in \mathcal{N}$ while ensuring that the total switching cost satisfies $C_i(T) := \sum_{t=1}^{T-1} c_{i,\bar{A}_i(t),\bar{A}_i(t+1)} \leq S$.

# 4 UNIT SWITCHING COST

We first consider the unit switching cost setting, where each player $p_i$ incurs a unit cost $c_{i,j,j'} = 1$ when switching from arm $a_j$ to another arm $a_{j'}$. In this setting, the switching cost budget $S$ represents the maximum number of switches allowable during interaction.

To avoid redundant operations and minimize unnecessary switching costs, our algorithm departs from the standard round-robin scheme by intensively pulling the same arm over a designated interval. Specifically, we partition the overall time horizon into multiple phases and, within each phase, allocate equal-length segments for each player to explore each arm exhaustively. After each player estimates their accuracy-based preference ranking over the arms, we invoke the Gale–Shapley algorithm (Gale & Shapley, 1962) to compute a player-optimal stable matching.

Consistent with prior work (Basu et al., 2021; Kong & Li, 2023; 2024), the algorithm begins with an index estimation phase lasting $N$ rounds. In this phase, each player repeatedly selects a specified arm $a_1$ until being accepted, after which the player withdraws from further selections. This procedure assigns each player a unique index which effectively mitigates conflicts in subsequent rounds.

Constrained by the switching budget $S$, the algorithm divides the remaining horizon into $q(S, K) = \lfloor(S-1)/(K-1)\rfloor$ phases. Within each phase, the players can switch between arms at most $K$ times, ensuring the opportunity to explore up to $K$ distinct arms. The selection of the phase length would be discussed later. For convenience, denote the starting round of the $\ell$-th phase as $t_\ell$, and its ending round would be $t_{\ell+1} - 1$. During each phase $\ell \in [q(S, K)]$, the players would uniformly explore the $K$ arms. Specifically, each arm is explored consecutively for a fixed number of $\lfloor(t_{\ell+1} - t_\ell)/K\rfloor$ rounds. To prevent conflicts between players, the function *ArmSequencer*() returns the overall arm order sequences as $\mathcal{T} = \{\tau_1, \tau_2, \ldots, \tau_N\}$, where

$$\tau_i = \left(a_{\text{Index}_i\%K+1}, a_{(\text{Index}_i+1)\%K+1}, \ldots, a_{(\text{Index}_i+K)\%K+1}\right)$$

is the round-robin order for each player $p_i$ to follow.

At the end of each phase, players would check whether they had estimated accurate preference rankings. Specifically, each player $p_i$ estimates an empirical preference value $\hat{\mu}_{i,j}$ for arm $a_j$ based on received rewards. It also maintains a confidence interval with lower bound LCB and upper bound UCB defined as $\text{LCB}_{i,j} = \hat{\mu}_{i,j} - \sqrt{6\log T/T_{i,j}}$ and $\text{UCB}_{i,j} = \hat{\mu}_{i,j} + \sqrt{6\log T/T_{i,j}}$, where $T_{i,j}$ is the number of matched times between player $p_i$ and arm $a_j$. If the confidence intervals of two arms are disjoint, the player can reliably identify the preference between them. Each player also verifies whether other players have successfully identified their top-$N$ ranked arms. If yes, starting from the next phase, each player can consistently select their most preferred arm and proceed to the following most preferred one among the remaining options once rejected (Gale & Shapley, 1962).

The selection of phase length plays a crucial role in the regret and switching cost analysis. A straightforward strategy of using equally-sized phases presents a dilemma: longer phases can incur excessive regret, while shorter phases result in a high number of switches. To address this trade-off, we adopt a growing phase size design. Early in the horizon, the phase size is small, allowing the player to make timely decisions once sufficient samples are collected. As the horizon progresses, the phase size increases, which helps reduce switching costs in subsequent rounds, thereby ensuring both low regret and minimal switching costs.

Our general framework, Algorithm 1, is designed to implement this principle, allowing for flexible adjustment to different trade-offs by tuning the growth rate of the phase size. We first propose a

geometric instantiation, where the time index of the exploration phase $u_l$ grow as $u_l = T^{1/M} u_{l-1}$ with $u_1 = T^{1/M}$. To further reduce switching costs, we also explore a faster growth rate—specifically, a minimax instantiation with $u_l = T^{1/(2-2^{1-M})} \sqrt{u_{l-1}}$ and $u_1 = T^{1/(2-2^{1-M})}$. In the extreme case, where only constant switching is allowed, we consider a single-phase instantiation, meaning there is only one phase for exploration. Since determining the optimal size of this single phase to collect appropriate samples is challenging, we invoke an oracle $\mathbb{O}$ to compute an approximation of a stable matching (Lin et al., 2024) after this phase. Please refer to the Appendix A.2 for a detailed description of the approximation oracle $\mathbb{O}$.

---

**Algorithm 1** Switching-Cost-Aware Matching (SCAM, abstract version from view of player $p_i$)

---

1: **Input:** Player set $\mathcal{N}$, arm set $\mathcal{K}$, horizon $T$, number of exploration phases $M$, time grid $\{t_l\}_{l=0}^{M+1}$
2: **Preprocessing:** $\tau_i \leftarrow$ *ArmSequencer*() `# Get exploration sequence for each player`
3: Initialize: $\hat{\mu}_{i,j} = 0, T_{i,j} = 0, \forall i \in \mathcal{N}, j \in \mathcal{K}$
4: $(i, \text{index}_i) \leftarrow$ *Index Estimation*(). `# Part 1, assign each player p_i with distinct index`
5: **for** $t = 1, 2, \ldots, T$ **do**
6:     `# Part 2, learn the preferences`
7:     **for** $l = 1$ to $M$ **do**
8:         Explore each arm $a \in \tau_i$ for $\lfloor \frac{(t_{l+1}-t_l)}{K} \rfloor$ *consecutive* rounds, updating its UCB and LCB.
9:         **if** CIs for all player's top $N + 1$ ranked arms are disjoint **then**
10:             $\mathcal{M}_{\text{final}} \leftarrow$ Gale-Shapley algorithm.
11:         **end if**
12:     **end for**
13:     `# Part 3, find the final matching and exploit`
14:     **if** exist player such that top $N + 1$ arms' CIs are not disjoint **then**
15:         $\mathcal{M}_{\text{final}} \leftarrow \mathbb{O}$.
16:     **end if**
17:     $M_t \leftarrow \mathcal{M}_{\text{final}}$.     `# Exploit the final matching`
18: **end for**

---

### 4.1 THEORETICAL ANALYSIS

Before presenting the theoretical results for three instantiations of Algorithm 1, we introduce some definitions that are crucial to our analysis. Definition 1 formalizes the minimum preference gap between the top $N + 1$ arms across all players.

**Definition 1** (Minimum Preference Gap). *Let $\rho_i = (\rho_{i,1}, \rho_{i,2}, \ldots, \rho_{i,K})$ denote the preference list of player $p_i$ over the arms, ordered from most to least preferred according to the true mean rewards $\mu_{i,j}$. Define $\Delta = \min_{i \in [K]; k, k' \in [N+1]} |\mu_{i,\rho_{i,k}} - \mu_{i,\rho_{i,k'}}|$ as the minimum preference gap, represents the smallest difference in mean rewards between any two of the top $N + 1$ arms for any player.*

Following Definition 2 introduces the concept of $\alpha$-fraction regret, a performance metric tailored for scenarios that rely on the approximation oracle $\mathbb{O}$.

**Definition 2** ($\alpha$-Fraction Regret). *The $\alpha$-fraction regret for player $p_i$ over time horizon $T$ is*

$$Reg_i^\alpha(T) = \alpha T \cdot \mu_{i,m^*} - \mathbb{E}\left[\sum_{t=1}^T X_i(t)\right],$$

*which is the difference between a fraction $\alpha$ of the total optimal reward and the expected cumulative reward obtained by the player.*

In the following, we provide the detailed regret and switching cost guarantees for Algorithm 1.

**Theorem 4.1.** *Under the unit switching cost setting, following Algorithm 1, the regret and switching cost satisfy*

*(a) With the geometric instantiation, the player-optimal stable regret for each player $p_i$ satisfies*

$$Reg_i(T) \leq O(T^{\frac{1}{q(S,K)}} \frac{K \log T}{\Delta^2}).$$

*This implies that $K \log T$ switches are sufficient to achieve a $O(K \log T / \Delta^2)$ regret.*

*(b) With the minimax instantiation, the player-optimal stable regret for each player $p_i$ satisfies*

$$Reg_i(T) \le O(T^{\frac{1}{2-2^{1-q(S,K)}}} \sqrt{\frac{K \log T}{\Delta^2}}).$$

*This implies that $K \log \log T$ switches can achieve $O(\sqrt{KT \log T/\Delta^2})$ regret.*

*(c) With the single-phase instantiation and phase length $T_0 = T^{2/3}(K \log T)^{1/3}$, the $\alpha$-fraction regret for each player $p_i$ satisfies*

$$Reg_i^\alpha(T) \le O((K \log T)^{\frac{1}{3}} T^{\frac{2}{3}}).$$

*$\alpha$ here is a ratio that sets a tractable benchmark for regret when the overlapping of preference estimates makes the true optimum unattainable. This instantiation only takes $2K$ switches.*

**Remark 4.1.** *The switching costs stated in Theorem 4.1(a) and Theorem 4.1(b) represent worst-case bounds, derived under the assumptions that the exploration part completes the maximum of $M = q(S, K)$ phases. In expectation, the algorithm can preform even better with lower switching costs of $O(\log \log T/\Delta^2))$ for the geometric and $O(\log \log(\log T/\Delta^2))$ for the minimax instantiation, please refer to Appendix C.4 for detailed explanation.*

## 5 GENERAL SWITCHING COST AND PARALLEL TRAVELING SALESMAN PROBLEM

To capture more realistic market dynamics, we extend our framework to the general switching cost setting. In this model, the cost incurred by a player $p_i$ when switching from arm $a_j$ to arm $a_{j'}$, denoted $c_{i,j,j'}$, is fully generalized. It depends on both the specific pair of arms and the player making the transition. Recall that a central component of our framework is *ArmSequencer*() introduced in Algorithm 1, which computes the cost-minimal exploration sequence for each player while ensuring no conflicts. The heterogeneity of switching costs presents a key challenge in designing efficient exploration sequences, which need to guarantee both minimal switching costs and no conflicts.

For convenience, we formalize this problem via a complete weighted graph $G = (\mathcal{K}, E, \{c_i\}_{i=1}^N)$, where the vertex set $\mathcal{K}$ represents the arms, the edge set $E$ encodes all possible switches, and the edge weights $c_i = \{c_{i,j,j'} : j, j' \in \mathcal{K}\}$ specify player-specific transition costs. The cumulative cost of an exploration sequence is then the sum of the edge weights along the path traversed by the player. It is worth noting that finding a sequence to minimize a single player's switching cost is equivalent to the classic Traveling Salesman Problem (Dantzig et al., 1954).

In the multi-player setting, to ensure collision-free exploration, a simple method is to further divide each phase into $N$ segments, where each player explores according to its least-cost switching order within its designated segment. We refer to this strategy as *Temporally Partitioned Exploration (TPE)*, and provide the detailed pseudocode in Algorithm 5 of Appendix A.3. In TPE, *ArmSequencer*() first determines the optimal tour for each player offline. Each exploration phase is then divided into $N$ non-overlapping segments, each assigned to a single player. Within its segment, a player sequentially traverses all $K$ arms along its cost-minimal path, while the other players remain idle. This design avoids collisions and ensures that each player follows its own cost-efficient trajectory, thereby minimizing switching costs. The drawback, however, is that each player is restricted to only $1/N$ of the phase duration, reducing its effective sample size and inflating its regret by a factor of $N$ relative to simultaneous exploration.

The TPE algorithm underscores a key trade-off: can we design a collision-free assignment that achieves low switching costs without incurring this $N$-fold penalty on regret? To address this question, we consider parallel exploration. If all $N$ players traverse the $K$ arms synchronously in a step-by-step manner, a full exploration can be completed in only $K$ steps, with each player finally returning to its starting arm to realize the closing cost of the Hamiltonian circuit. Crucially, the market constraint—that no two players may occupy the same arm in the same round—must be strictly satisfied. This requirement motivates a more sophisticated problem formulation.

### 5.1 PARALLEL TRAVELING SALESMAN PROBLEM

To achieve efficient, simultaneous exploration, we frame the task for *ArmSequencer*() as finding cost-minimal, collision-free paths by solving the *Parallel Traveling Salesman Problem (PTSP)*. This

problem seeks to coordinate the exploration sequences of all players to minimize their total switching cost while respecting the structural constraints of the matching market.

### 5.1.1 PROBLEM FORMULATION

The *Parallel Traveling Salesman Problem (PTSP)* is defined over the player-specific cost graph $G = (\mathcal{K}, E, \{c_i\}_{i=1}^N)$. The objective is to construct a collection of $N$ Hamiltonian circuits, $\mathcal{T} = \{\tau_1, \tau_2, \ldots, \tau_N\}$, where each circuit $\tau_i = (\tau_i(1), \tau_i(2), \ldots, \tau_i(K), \tau_i(1))$ specifies the traversal order of all arms by player $p_i$. The cost of a circuit $\tau_i$ is defined as $C_i(\tau_i) = \sum_{t=1}^{K-1} c_{i,\tau_i(t),\tau_i(t+1)} + c_{i,\tau_i(K),\tau_i(1)}$. The global objective is to minimize the total cost across all players:

$$\min_{\mathcal{T}} \quad \sum_{i=1}^N C_i(\tau_i)$$

$$\text{s.t.} \quad \bigcup_{t=1}^K \tau_i(t) = \mathcal{K}, \quad \forall\, i \in \mathcal{N}, \tag{1}$$

$$\tau_i(t) \neq \tau_{i'}(t), \quad \forall\, t \in \{1, \ldots, K\}, \forall\, i \neq i'. \tag{2}$$

Constraints (1) and (2) capture the two fundamental requirements of PTSP. Constraint (1) enforces exhaustive traversal: each player's circuit must visit every arm exactly once before returning to its starting point, thereby ensuring full coverage of the arm set $\mathcal{K}$. Constraint (2) imposes collision avoidance, requiring that no two players occupy the same arm simultaneously. This formulation is NP-hard, as it generalizes the classic Traveling Salesman Problem (TSP) (Dantzig et al., 1954) to a multi-player, constrained setting.

### 5.1.2 HEURISTIC ALGORITHM FOR PARALLEL TRAVELING SALESMAN PROBLEM

To address the NP-hardness of PTSP, we develop heuristic algorithms executed by *ArmSequencer*() as an offline preprocessing step. As a natural starting point, one may let each player independently solve its own TSP instance, thereby obtaining a Hamiltonian circuit $\tau_i^*$ for every player $p_i$. If these tours can be jointly executed without collisions, they already yield a feasible solution. Importantly, the notion of collision here is more general than the instantaneous conflicts captured by Constraint (2). Specifically, even if two players' tours overlap in space, conflicts may still be avoided if their starting points along the cycles are suitably offset. Thus, the collision detection problem is to decide whether there exists a configuration of starting offsets for all players such that, over the entire execution, no two players ever occupy the same arm simultaneously. We defer the formal definition of this offset-assignment feasibility problem to Appendix B. Whenever a collision-free configuration exists, we directly adopt the independent-TSP solution (part 1 in Algorithm 2).

In the general case, however, independently computed tours are likely to conflict. To robustly enforce collision avoidance, we then call back to the *Aggregated Cost Heuristic (ACH)*, described in part 2 of Algorithm 2. Rather than optimizing separately, ACH operates on the collective system cost by constructing a single aggregated graph $\bar{G}$. For each edge $(j, j')$, the aggregated weight is defined as the sum of player-specific costs, $\bar{c}_{j,j'} = \sum_{i=1}^N c_{i,j,j'}$. We then compute a single shortest Hamiltonian circuit $\bar{\tau}$ on $\bar{G}$. This common tour is assigned to all players, but with distinct rotated starting points. Players then follow the same relative order of arms while being phase-shifted, guaranteeing a collision-free execution.

Before presenting the theoretical guarantee for ACH, we introduce a mild structural assumption that ensures cost heterogeneity across players remains bounded.

**Assumption 5.1** (Heterogeneity Parameter). *For any edge $(j, j') \in E$ and any two players $i, i' \in \mathcal{N}$, the ratio of their edge costs is bounded by a constant $\gamma \geq 1$, i.e.,*

$$\max_{i,i' \in \mathcal{N}, j \neq j'} \frac{c_{i,j,j'}}{c_{i',j,j'}} \leq \gamma.$$

This assumption is practical motivated. While costs are player-dependent, they fundamentally reflect the intrinsic difficulty of the underlying transition between arms. Consequently, although costs for a given transition may differ across players, they remain within a comparable order of magnitude.

---

**Algorithm 2** Aggregated Cost Heuristic (ACH with Baseline Check)

---

1: **Input:** $G = (\mathcal{K}, E, \{c_i\}_{i=1}^N)$
2: **Output:** A set of collision-free Hamiltonian circuits $\mathcal{T}^* = \{\tau_1^*, \ldots, \tau_N^*\}$
3: `# Part 1, Independent − TSP Baseline`
4: **for** each player $i = 1, \ldots, N$ **do**
5:     $\tau_i^* \leftarrow TSP\_Solver(G, \text{cost} = c_i)$
6: **end for**
7: **if** $Collision\_Free(\mathcal{T}^*)$ **then**
8:     `# Refer to Appendix B for collision detection procedure`
9:     Return $\mathcal{T}^* = \{\tau_1^*, \ldots, \tau_N^*\}$
10: **end if**
11: `# Part 2, Aggregated Cost Heuristic`
12: Initialize a new graph $\bar{G} = (\mathcal{K}, E, \bar{c})$.
13: **for** each edge $(j, j') \in E$ **do**
14:     $\bar{c}_{j,j'} \leftarrow \sum_{i=1}^N c_{i,j,j'}$. `# Sum the costs from all players for this edge`
15: **end for**
16: $\bar{\tau} \leftarrow TSP\_Solver(\bar{G})$. `# Find the shortest Hamiltonian circuit`
17: Let $\bar{\tau} = (\bar{\tau}(1), \ldots, \bar{\tau}(K), \bar{\tau}(1))$.
18: Initialize an empty set of circuits $\mathcal{T}$.
19: **for** player index $i = 1, \ldots, N$ **do**
20:     `# Assign a unique starting point by rotating the common path`
21:     $\tau_i^* \leftarrow (\bar{\tau}(i), \bar{\tau}(i+1), \ldots, \bar{\tau}(K), \bar{\tau}(1), \ldots, \bar{\tau}(i-1), \bar{\tau}(i))$.    `# Indices are modulo K`
22:     Add $\tau_i^*$ to $\mathcal{T}^*$.
23: **end for**
24: **return** $\mathcal{T}^*$

---

**Theorem 5.1.** *Let $\mathcal{T}^*$ be the set of collision-free Hamiltonian circuits returned by the Aggregated Cost Heuristic (Algorithm 2). Let ALG denotes the total cost of this solution, and $OPT_{par}$ be the cost of an optimal solution to the Parallel Traveling Salesman Problem. Under Assumption 5.1, the total cost of the Aggregated Cost Heuristic solution satisfies:*

$$ALG \leq \gamma \cdot \beta \cdot OPT_{par}$$

*where $\beta$ is the approximation factor of the underlying TSP algorithm (e.g., $\beta = 1.5$ for Christofides-Serdyukov (Christofides, 2022)) and $\gamma$ is the heterogeneity parameter.*

**Remark 5.1.** *Theorem 5.1 shows that ACH inherits the approximation ratio of the base TSP solver up to a multiplicative factor $\gamma$ accounting for player heterogeneity. In particular, when costs are nearly homogeneous across players ($\gamma \approx 1$), ACH provides a constant-factor approximation to the optimal PTSP solution. This highlights ACH as an effective and theoretically grounded heuristic for collision-free multi-agent exploration under general switching costs. Please refer to Appendix C.5 for the proof of Theorem 5.1.*

In this section, with the general switching cost setting, the number of exploration phases is given by $M = \lfloor (S - \max_{i,j,j'} c_{i,j,j'})/\text{ALG} \rfloor$. Under Algorithm 5 with geometric instantiation, the regret incurs an additional $N$-fold penalty, and the exponential horizon factor is approximately $T^{(\beta \cdot \text{OPT})/S}$, where OPT is the sum of the players' optimal TSP costs. Under Algorithm 1, using Algorithm 2 for the offline part, this factor is $T^{(\beta \cdot \text{OPT}_{\text{par}})/S}$ if the baseline check succeeds, and $T^{(\gamma \cdot \beta \cdot \text{OPT}_{\text{par}})/S}$ otherwise. The trade-offs are evident: Algorithm 5 attains lower costs but suffers an $N$-fold penalty, while the baseline avoids this penalty at the expense of larger $T$-factors. When baseline check fails, ACH also eliminates the $N$-fold penalty but adds a $\gamma$ factor, showing that solution quality directly governs the tightness of the regret bound.

## 6 CONCLUSION

We introduced a switching-cost-aware framework for bandit learning in matching markets and analyzed its complexity via a PTSP formulation. Our algorithms balance regret and switching costs, with heuristics that are both efficient and provably near-optimal. This work opens avenues for studying richer markets with dynamic or unknown preferences.

## 7 ETHICS STATEMENT

This work is purely theoretical and does not involve human subjects, personal data, or sensitive attributes. The algorithms are studied in the context of matching markets and analyzed through synthetic simulations under well-specified assumptions. Potential risks mainly relate to downstream applications: in real-world platforms such as online labor markets, automated decision-making could introduce fairness or accessibility concerns if applied without safeguards. However, our study is intended as a methodological contribution and does not directly deploy models in sensitive domains. We encourage practitioners to carefully consider ethical implications, such as fairness, bias, and transparency, when adapting our algorithms to practical systems.

## 8 REPRODUCIBILITY STATEMENT

We have taken several steps to ensure the reproducibility of our results. All theoretical analyses, including regret bounds and switching-cost trade-offs, are provided with detailed proofs in the appendix (see Sections C and D). The algorithms are described with full pseudocode (Algorithms 1–5 in the main text and appendix) , and their assumptions are explicitly stated. Experimental evaluations are based on synthetic data with controlled preference structures, and we specify all parameter settings such as horizon length, number of players, and cost functions. Upon acceptance, we will release our implementation, including code for simulations and evaluation scripts, to further support replication and extension of our work.

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

# A ALGORITHMS

## A.1 SWITCHING-COST-AWARE MATCHING

Here we provide the detailed version of Switching-Cost-Aware Matching in Algorithm 3. Note that during the entire online process, the start time of phase $l$ is $t = (N + u_{l-1} + l - 1) + 1$, where $l - 1$ denotes the monitoring rounds in previous phases and $N$ denotes the index estimation phase.

---

**Algorithm 3** Switching-Cost-Aware Matching (SCAM, abstract version from view of player $p_i$)

---

1: **Input:** Player set $\mathcal{N}$, arm set $\mathcal{K}$, horizon $T$, number of exploration phases $M + 1$, time grid $\{t_l\}_{l=0}^{M}$, arm order sequencing function *ArmSequencer*()
2: Initialize: $\hat{\mu}_{i,j} = 0, T_{i,j} = 0, \forall i \in \mathcal{N}, j \in \mathcal{K}$. `use_oracle` $\leftarrow$ `True`, $F_l \leftarrow$ `False`
3: # `Part 1, index estimation`
4: Arm $\leftarrow a_1$
5: **for** $t = 1, 2, \ldots, N$ **do**
6:    $A_i(t) \leftarrow$ Arm
7:    **if** $\bar{A}_i(t) = a_1$ **then**
8:       Index $\leftarrow t$;   Arm$\leftarrow a_2$
9:    **end if**
10: **end for**
11: $\tau_i \leftarrow$ *ArmSequencer*() #
12: # `Part 2, preference estimation`
13: **for** $l = 1$ to $M$ **do**
14:    **for** each arm $a \in \tau_i$ **do**
15:       Choose arm $a$ for $\lfloor \frac{(t_{l+1} - t_l)}{K} \rfloor$ *consecutive* rounds within phase $l$
16:       Observe $X_{i,A_i(t)}(t)$ and update $\hat{\mu}_{i,A_i(t)}, T_{i,A_i(t)}$ if matched
17:    **end for**
18:    Compute $\text{UCB}_{i,j}$ and $\text{LCB}_{i,j}$ for each $j \in [K]$ #
19:    **if** $\exists \sigma$ such that $\text{LCB}_{i,\sigma_k} > \text{UCB}_{i,\sigma_k}$ for any $k \in [N]$
        and $\text{LCB}_{i,\sigma_N} > \text{UCB}_{i,\sigma_k}$ for any $k = N + 1, N + 2, \ldots, K$ **then**
20:       $F_l =$ `True` and $\sigma_i = \sigma$
21:    **end if**
22:    At $t = (N + u_l + l - 1) + 1$ # `Monitoring round (decentralized check)`
23:    Initiatlize $\mathcal{O} \leftarrow \emptyset$
24:    **if** $F_i =$ `True` **then**
25:       Propose to arm $a_{\text{Index}}$ in $\sigma_i$, $A_i(t) = a_{\text{Index}}$
26:    **else**
27:       Stay idle, $A_i(t) = \emptyset$
28:    **end if**
29:    Update $\mathcal{O} = \cup_{i=1}^{N} \{\bar{A}_i(t)\}$
30:    **if** $\mathcal{O} == N$ **then**
31:       Enter next part with $\sigma_i; t_2 = t$
32:       **break** exploration loop.
33:    **end if**
34:    **if** $l == M$ and $\mathcal{O} == N$ **then**
35:       Enter part 4 with $\sigma_i$; $t_2 = t$
36:       `use_oracle` $\leftarrow$ `True`
37:    **end if**
38: **end for**
39: # `Part 3, find final matching and exploit`
40: **if** `use_oracle` is `True` **then**
41:    $\mathcal{M}_{\text{final}} \leftarrow \mathbb{O}$
42:    **for** $t = t_2 + 1, t_2 + 2, \ldots$ **do**
43:       $M_t = \mathcal{M}_{\text{final}}$
44:    **end for**
45: **else**
46:    #Implement Gale $-$ Shapley algorithm
47:    Initialize $s = 1$
48:    **for** $t = t_2 + 1, t_2 + 2, \ldots$ **do**
49:       $A_i(t) = a_{\sigma_{i,s}}$
50:       $s = s + 1$ if $\bar{A}_i(t) == \emptyset$
51:    **end for**
52: **end if**

---

## A.2 Approximation Oracle $\mathbb{O}$

Following the framework of Lin et al. (2024), we present the approximation oracle $\mathbb{O}$ used in our main algorithm. This oracle is invoked as a fallback mechanism when the exploration phase concludes without all players being able to resolve their preference rankings—that is, when confidence intervals for some arms' estimated rewards still overlap. This appendix provides a self-contained description of $\mathbb{O}$, adapting the relevant definitions and guarantees from Lin et al. (2024) to the notation used in this paper.

### A.2.1 Robustness to Estimation Uncertainty

Since the oracle operates on empirical reward estimates rather than true mean rewards, we require a notion of stability that is robust to small estimation errors. This is captured by $\epsilon$-stability, which relaxes the standard stability condition.

**Definition 3** ($\epsilon$-stability). *Let $\epsilon \geq 0$. A matching $\mathcal{M}$ is called $\epsilon$-stable if there is no blocking pair $(p_i, a_k)$ such that arm $a_k$ strictly prefers player $p_i$ to its current match $\mathcal{M}(k)$, and player $p_i$'s preference satisfies $\mu_{i,k} > \mu_{i,\mathcal{M}(i)} + \epsilon$.*

This relaxation allows for blocking pairs only if the utility gain for the player exceeds a tolerance $\epsilon$, making the concept suitable for noisy, empirically estimated preferences. We can now define the performance guarantee of our oracle with respect to this robust stability concept.

### A.2.2 Oracle Implementation and Guarantee

The key idea behind the oracle, presented in Algorithm 4, is to transform the problem with overlapping (tied) preferences into one with strict preferences by creating multiple "duplicates" of each arm. The utilities for these duplicates are slightly penalized, which effectively breaks the ties. A standard Gale-Shapley algorithm is then run on this larger, duplicated market, and the resulting matching is projected back to the original set of arms. By randomizing over these projections, the oracle provides the desired approximation guarantee.

**Theorem A.1** (Oracle Guarantee, adapted from Lin et al. (2024)). *Let $m = \lceil \log_2 N \rceil$ and $\epsilon \geq 0$. Algorithm 4 outputs a distribution $D$ over matchings such that for every player $p_i \in \mathcal{N}$,*

$$\mathbb{E}_{\mathcal{M} \sim D}\left[\hat{\mu}_{i,\mathcal{M}(i)}\right] \geq \frac{1}{m}\mu_i^* - \epsilon,$$

*where $\mu_i^*$ denotes the optimal $\epsilon$-stable payoff for player $i$.*

For clarity in Algorithm 4, let $a_k(r)$ denote the $r$-th duplicate of arm $a_k \in \mathcal{K}$ for $r \in \{1, \dots, m\}$. The adjusted utility for this duplicate is defined as $\hat{\mu}_{i,k(r)} := \hat{\mu}_{i,k} - (r-1)\epsilon$. Players' preferences are then induced by these adjusted utilities, with any remaining ties broken in favor of a smaller duplicate index $r$. The arms' original preference rankings over players are copied to all of their respective duplicates.

---

**Algorithm 4** $\epsilon$-Oracle for Approximation Stable Matching

---

1: **Input:** Player set $\mathcal{N}$, arm set $\mathcal{K}$, estimated mean rewards $\{\hat{\mu}_{i,k}\}$, duplication factor $m = \lceil \log_2 N \rceil$, tolerance $\epsilon \geq 0$
2: For each arm $a_k \in \mathcal{K}$, create $m$ duplicates: $a_k(1), \dots, a_k(m)$.
3: For each player $p_i \in \mathcal{N}$ and duplicate $a_k(r)$, set adjusted utility

$$\hat{\mu}_{i,k(r)} \leftarrow \hat{\mu}_{i,k} - (r-1)\epsilon.$$

   Break ties in favour of smaller $r$.
4: Run the Gale-Shapley algorithm with players proposing to the set of duplicated arms to find a player-optimal stable matching $\tilde{\mathcal{M}}$.
5: For each $r \in \{1, \dots, m\}$, project $\tilde{\mathcal{M}}$ into the original arms by defining a matching $\mathcal{M}_r$, where $\mathcal{M}_r$ is the original arm corresponding to player $i$'s partner in $\tilde{\mathcal{M}}$ from the $r$-th set of duplicates.
6: **Return** $D$, the uniform distrbution over $\{\mathcal{M}_1, \dots, \mathcal{M}_m\}$

---

The logarithmic number of duplications, $m = \lceil \log_2 N \rceil$, ensures the algorithm runs in polynomial time. Penalizing duplicates by increments of $\epsilon$ guarantees that the output satisfies the $\epsilon$-stability property, while randomization over the projections provides the approximation factor. This makes the oracle a theoretically sound and efficient fallback for our main algorithm.

**Definition 4** (($\alpha, \epsilon$)-Approximation Oracle). *An oracle $\mathbb{O}$ is an ($\alpha, \epsilon$)-approximation oracle if, given estimated mean rewards $\{\hat{\mu}_{i,k}\}$, it outputs a (possibly randomized) matching $\tilde{\mathcal{M}}$ such that for every player $p_i \in \mathcal{N}$,*

$$\mathbb{E}\big[\hat{\mu}_{i,\tilde{\mathcal{M}}(i)}\big] \geq \alpha \cdot \mu_i^* - \epsilon,$$

*where $\mu_i^*$ denotes the maximum payoff player $i$ can achieve across all possible $\epsilon$-stable matchings. The expectation is taken over the oracle's internal randomness. Here, $\alpha \in (0,1]$ is the multiplicative approximation ratio, and $\epsilon$ is the stability tolerance.*

The Algorithm 4 ensures that for any estimated mean rewards $\{\hat{\mu}_{i,k}\}$ input, $\alpha \geq 1/\log_2 N$. This definition formalizes the guarantee we require: in expectation, each player secures at least an $\alpha$-fraction of their best possible payoff in any $\epsilon$-stable matching, minus a small additive error $\epsilon$.

### A.3 TEMPORALLY PARTITIONED EXPLORATION (TPE)

---

**Algorithm 5** Temporally Partitioned Exploration (TPE)

---

1: **Input:** Player set $\mathcal{N}$, arm set $\mathcal{K}$, horizon $T$, number of exploration phases $M$, time grid $\{t_l\}_{l=0}^{M+1}$
2: **Preprocessing:** $\mathcal{T} \leftarrow ArmSeqencer(G)$ # Returns a set of shortest Hamiltonian paths $\{\tau_1, \ldots, \tau_N\}$
3: # Initialization and part 1 are the same as Algorithm 1
4: **for** each exploration phase $l = 1, \ldots, M$ **do**
5: $\quad D_l = t_{l+1} - t_l$. # Calculate the duration of each exploration phase
6: $\quad$ Partition the phase into $N$ segments, each of duration $\frac{D_l}{N}$.
7: $\quad$ **for** each player $p_i \in \mathcal{N}$ **do**
8: $\quad\quad$ **wait** until segment $i$ begins.
9: $\quad\quad$ **for** each arm $a_j \in \tau_i$ **do**
10: $\quad\quad\quad$ Player $p_i$ pulls arm $a_j$ for $\left\lfloor \frac{D_l}{NK} \right\rfloor$ *consecutive* rounds.
11: $\quad\quad$ **end for**
12: $\quad\quad$ Player $p_i$ remains idle until the next phase.
13: $\quad$ **end for**
14: $\quad$ **if** CIs for all player's top $N + 1$ ranked arms are disjoint **then**
15: $\quad\quad$ $\mathcal{M}_{\text{final}} \leftarrow$ Gale-Shapley algorithm.
16: $\quad$ **end if**
17: **end for**
18: # Part 3 is the same as Algorithm 1

---

## B COLLISION DETECTION VIA OFFSET ASSIGNMENT

In Section 5.1.2, our heuristic algorithm for the PTSP begins with a baseline check: if the collection of individually Hamiltonian circuits under a specific TSP solver is already collision-free, then it constitutes a strong solution. This appendix formalizes the associated decision problem of determining whether a given set of tours can be rendered collision-free by adjusting their starting points.

Even when two tours, $\tau_i^*$ and $\tau_{i'}^*$, overlap on certain arms, collisions can still be avoided if the players traverse them with appropriate temporal offsets. For example, if all players were to follow the identical tour, assigning each a distinct starting offset would trivially prevent collisions. This naturally leads to a general feasibility problem.

### B.1 PROBLEM FORMULATION

The core question is whether there exists a set of relative phase shifts that guarantees collision-free execution. This can be viewed as a circular scheduling or cyclic shift assignment problem. Formally, given $N$ Hamiltonian circuits $\mathcal{T}^* = \{\tau_1^*, \ldots, \tau_N^*\}$, the objective is to find offsets $S = \{s_1, \ldots, s_N\}$ with $s_i \in \{1, \ldots, K\}$ such that, for all distinct $i, i' \in \mathcal{N}$ and for all $t \in \{1, \ldots, K\}$, the collision-avoidance constraint holds:

$$\tau_i^* \left( (t + s_i) \bmod K \right) \neq \tau_{i'}^* \left( (t + s_{i'}) \bmod K \right).$$

The decision problem is thus to determine whether a valid assignment $S$ exists.

### B.2 SOLUTION APPROACHES

Determining feasibility is computationally challenging. We highlight two representative approaches:

- Brute-Force Search: Exhaustively enumerate all $K^N$ possible offset assignments. While conceptually simple, this approach is only practical for very small $N$ and $K$ due to its exponential complexity.

- Constraint Satisfaction Formulation: The problem can be cast as a discrete CSP, where the variables are the offsets $\{s_i\}_{i=1}^N$, each with domain $\{1, \ldots, K\}$, and the constraints are the pairwise inequalities defined above. This formulation enables the use of general-purpose CSP solvers, such as SAT or Integer Linear Programming (ILP) solvers, to test feasibility.

If a feasible offset assignment is found, it confirms that the independently optimized tours can be executed in parallel without collisions. Otherwise, a fallback mechanism such as the Aggregated Cost Heuristic (ACH) must be employed to enforce collision avoidance by design.

## C PROOF

### C.1 PROOF OF THEOREM 4.1(A)

For convenience, denote by $\hat{\mu}_{i,j}(t)$, $T_{i,j}(t)$, $\mathrm{UCB}_{i,j}(t)$ and $\mathrm{LCB}_{i,j}(t)$ the values of $\hat{\mu}_{i,j}$, $T_{i,j}$, $\mathrm{UCB}_{i,j}$ and $\mathrm{LCB}_{i,j}$ at the end of round $t$, respectively. Define the bad event

$$\mathcal{F} = \left\{ \exists t \in [T], i \in [N], j \in [K] : |\hat{\mu}_{i,j}(t) - \mu_{i,j}| > \sqrt{\frac{6 \log T}{T_{i,j}(t)}} \right\},$$

i.e., that at some point an arm's true mean lies outside its estimated confidence interval over the horizon $T$. because in each phase all players sample arms in lockstep and collect the same amount of data, every player enters part 3 simultaneously (see Line 31 of Algorithm 3). Let $l_{\max}$ be the final phase index at which part 2 completes and part 3 begins. Equivalently, by time $t_{l_{\max}}$ every player has correctly recovered their total ranking of the arms. We can then state the following regret bound.

The optimal stable regret of each player $p_i$ by following algorithm 1 satisfies

$$
\begin{aligned}
Reg_i(T) &= \mathbb{E}\left[ \sum_{t=1}^{T} (\mu_{i,m_i^*} - X_i(t)) \right] \\[2mm]
&\leq \mathbb{E}\left[ \sum_{t=1}^{T} \mathbb{1}\{\overline{A}(t) \neq m^*\} \cdot \Delta_{i,\max} \right] \\[2mm]
&\leq N\Delta_{i,\max} + \mathbb{E}\left[ \sum_{t=N+1}^{t} \mathbb{1}\{\overline{A}(t) \neq m^*\} \,|\, \neg\mathcal{F} \right] \cdot \Delta_{i,\max} + \mathbb{P}(\mathcal{F}) \cdot T \cdot \Delta_{i,\max} \\[2mm]
&\leq N\Delta_{i,\max} + \mathbb{E}\left[ \sum_{t=N+1}^{t} \mathbb{1}\{\overline{A}(t) \neq m^*\} \,|\, \neg\mathcal{F} \right] \cdot \Delta_{i,\max} + 2NK\Delta_{i,\max} \qquad (3) \\[2mm]
&\leq N\Delta_{i,\max} + \mathbb{E}\left[ \sum_{l=1}^{l_{\max}} (T^{\frac{l}{q(S,K)}} + 1) + N^2 \,|\, \neg\mathcal{F} \right] \cdot \Delta_{i,\max} + 2NK\Delta_{i,\max} \qquad (4) \\[2mm]
&\leq N\Delta_{i,\max} + \left( T^{\frac{1}{q(S,K)}} \cdot (96K\mathrm{log}T/\Delta^2) + \frac{1}{\mathrm{log}T^{\frac{1}{q(S,K)}}} \mathrm{log}\left( (T^{\frac{1}{q(S,K)}} - 1) \cdot \frac{96K\mathrm{log}T}{\Delta^2} + T^{\frac{1}{q(S,K)}} \right) \right) \cdot \Delta_{i,\max}
\end{aligned}
$$

$$+ N^2 \Delta_{i,\max} + 2NK\Delta_{i,\max}. \tag{5}$$

where Eq.(3) comes from Lemma C.1, Eq.(4) holds according to Algorithm 3 and Lemma C.2 , Eq.(5) holds based on Lemma C.4.

**Lemma C.1.**
$$\mathbb{P}(\mathcal{F}) \leq 2NK/T.$$

*Proof.*

$$\mathbb{P}(\mathcal{F}) = \mathbb{P}\left(\exists 1 \leq t \leq T, i \in [N], j \in [K] : |\hat{\mu}_{i,j}(t) - \mu_{i,j}| > \sqrt{\frac{6\log T}{T_{i,j}(t)}}\right)$$

$$\leq \sum_{t=1}^{T} \sum_{i\in[N]} \sum_{j\in[K]} \mathbb{P}\left(|\hat{\mu}_{i,j}(t) - \mu_{i,j}| > \sqrt{\frac{6\log T}{T_{i,j}(t)}}\right)$$

$$\leq \sum_{t=1}^{T} \sum_{i\in[N]} \sum_{j\in[K]} \sum_{s=1}^{t} \mathbb{P}\left(T_{i,j}(t) = s, |\hat{\mu}_{i,j}(t) - \mu_{i,j}| > \sqrt{\frac{6\log T}{s}}\right)$$

$$\leq \sum_{t=1}^{T} \sum_{i\in[N]} \sum_{j\in[K]} t \cdot 2\exp(-3\ln T)$$

$$\leq 2NK/T,$$

the second last inequality comes from Lemma D.1.

$\square$

**Lemma C.2.** *Conditioned on $\neg\mathcal{F}$, phase 3 requires at most $N^2$ rounds before each player $p_i$ reaches her optimal stable arm $m_i^*$. Moreover, in all subsequent rounds $s$ is never updated and $p_i$ remains accepted by $m_i^*$.*

*Proof.* By Lemma C.5 and Algorithm 3, when player $p_i$ enters Phase 3 with ordering $\sigma_i$, her top-$N$ arms under $\sigma_i$ coincide exactly with her true top-$N$ preferences:

$$\mu_{i,\sigma_{i,k}} > \mu_{i,\sigma_{i,k+1}} \quad \forall k \in [N], \qquad \mu_{i,\sigma_{i,N}} > \mu_{i,\sigma_{i,k}} \quad \forall k = N+1, \ldots, K.$$

By Lemma C.4, all players begin Phase 3 simultaneously. Combining Lemma C.3, the procedure they follow is precisely the offline Gale–Shapley algorithm run on $N$ players and their top-$N$ arms. It is well known that Gale–Shapley terminates after at most $N^2$ proposals. Therefore Phase 3 also halts within $N^2$ rounds, at which point every player has secured her stable match $m_i^*$. From then on no further rejections occur, so $s$ remains fixed and $p_i$ is always accepted by $m_i^*$. $\square$

**Lemma C.3.** *In the offline GS algorithm, at most $N$ arms have been proposed by players before the algorithm stops, thus the player-optimal stable arm of each player must br its first $N$-ranked. And GS would reach player-optimal stability in at most $N^2$ steps.*

*Proof.* In the player-proposing Gale-Shapley algorithm with $N$ players and $K$ arms, each player starts by proposing to her top choice and, upon rejection, moves to the next arm in her preference list. If any player were ever to propose more than $N$ times, her final match would lie outside her top-$N$ arms. But then by pigeonhole at least one of her top-$N$ arms must be unmatched, creating a blocking pair and violating stability. Therefore no player can be rejected (and hence propose) more than $N$ times, so there are at most $N^2$ proposals in total, and the algorithm terminates within $N^2$ steps. $\square$

**Lemma C.4.** *Conditional on $\neg\mathcal{F}$, phase 2 will proceed in at most $l_{max}$ batches where*

$$l_{max} = \min\left\{l : b^l \geq 96K logT/\Delta^2\right\},$$

*which implies that $T^{\frac{l_{max}}{q(S,K)}} \leq T^{\frac{1}{q(S,K)}} \cdot \left(96K logT/\Delta^2 + 1\right)$ and $l_{max} \leq \frac{1}{logT^{\frac{1}{q(S,K)}}} log\left(T^{\frac{1}{q(S,K)}} \cdot \frac{96K logT}{\Delta^2}\right)$. And all players will enter phase 3 simultaneously at*

*time grid $l_{max}$.*

*Proof.* Since
$$T^{\frac{l_{\max}}{q(S,K)}} \geq 96K\log T/\Delta^2,$$
we have
$$T^{\frac{l_{\max}-1}{q(S,K)}} \leq 96K\log T/\Delta^2.$$
Then we get
$$T^{\frac{l_{\max}}{q(S,K)}} \leq 96T^{\frac{1}{q(S,K)}}K\log T/\Delta^2.$$

According to the summation formula of geometric series we get

$$l_{\max} \leq \frac{1}{\log T^{\frac{1}{q(S,K)}}} \log \left( T^{\frac{1}{q(S,K)}} \cdot \frac{96K\log\mathrm{T}}{\Delta^2} \right).$$

$\square$

**Lemma C.5.** *Conditional on $\neg\mathcal{F}$, $\mathrm{UCB}_{i,j}(t) < \mathrm{LCB}_{i,j'}(t)$ implies $\mu_{i,j} < \mu_{i,j'}$.*

*Proof.* According to the definition of LCB and UCB, conditional on $\neg\mathcal{F}$, we have

$$\mathrm{LCB}_{i,j}(t) = \hat{\mu}_{i,j}(t) - \sqrt{\frac{6\log T}{T_{i,j}(t)}} \leq \mu_{i,j} \leq \hat{\mu}_{i,j}(t) + \sqrt{\frac{6\log T}{T_{i,j}(t)}} = \mathrm{UCB}_{i,j}(t).$$

Thus if $\mathrm{UCB}_{i,j}(t) < \mathrm{LCB}_{i,j'}(t)$, there would be

$$\mu_{i,j} \leq \mathrm{UCB}_{i,j}(t) < \mathrm{LCB}_{i,j'}(t) < \mu_{i,j'}.$$

Thus the lemma is proved. $\square$

**Lemma C.6.** *In round $t$, let $T_i(t) = \min_{j \in [K]} T_{i,j}(t)$ and $\bar{T}_i = 96\log T/\Delta^2$. Conditional on $\neg\mathcal{F}$, if $T_i(t) > \bar{T}_i$, we have $\mathrm{LCB}_{i,\rho_{i,k}}(t) > \mathrm{UCB}_{i,\rho_{i,k+1}}(t)$ for any $k \in [N]$, and $\mathrm{LCB}_{i,\rho_{i,N}}(t) > \mathrm{UCB}_{i,\rho_{i,k}}(t)$ for any $k = N+1, N+2, \ldots, K$.*

*Proof.* Assume, toward contradiction, taht there exists some index $k$ (either $k \leq N$ or $k > N$) for which
$$\mathrm{LCB}_{i,\rho_{i,k}}(t) \leq \mathrm{UCB}_{i,\rho_{i,k+1}}(t).$$
Let $j$ be the arm realizing the right-hand side and $j'$ the arm realizing the left-hand side in this inequality. On the event $\neg\mathcal{F}$, we know simultaneously

$$\mu_{i,j'} - 2\sqrt{\frac{6\log T}{T_i(t)}} \leq \mathrm{LCB}_{i,j'}(t) \leq \mathrm{UCB}_{i,j}(t) \leq \mu_{i,j} + 2\sqrt{\frac{6\log T}{T_i(t)}}.$$

It follows that

$$\Delta_{i,j,j'} = \mu_{i,j'} - \mu_{i,j} \leq 4\sqrt{\frac{6\log T}{T_i(t)}} \implies T_i(t) \leq \frac{96\log T}{\Delta_{i,j,j'}^2} \leq \frac{96\log T}{\Delta^2} = \bar{T}_i,$$

contradicting the assumption $T_i(t) > \bar{T}_i$. Hence the claimed inequalities on the UCBs and LCB s must hold. $\square$

## C.2   PROOF OF THEOREM 4.1(B)

The optimal stable regret of each player $p_i$ under Algorithm 4.1(b) satisfies

$$Reg_i(T) = \mathbb{E}\left[\sum_{t=1}^{T}(\mu_{i,m_i^*} - X_i(t))\right]$$

$$\leq \mathbb{E}\left[\sum_{t=1}^{T}\mathbb{1}\{\overline{A}(t) \neq m^*\} \cdot \Delta_{i,\max}\right]$$

$$\leq N\Delta_{i,\max} + \mathbb{E}\left[\sum_{t=N+1}^{t}\mathbb{1}\{\overline{A}(t) \neq m^*\} \mid \neg\mathcal{F}\right] \cdot \Delta_{i,\max} + \mathbb{P}(\mathcal{F}) \cdot T \cdot \Delta_{i,\max}$$

$$\leq N\Delta_{i,\max} + \mathbb{E}\left[\sum_{t=N+1}^{t}\mathbb{1}\{\overline{A}(t) \neq m^*\} \mid \neg\mathcal{F}\right] \cdot \Delta_{i,\max} + 2NK\Delta_{i,\max} \tag{6}$$

$$\leq N\Delta_{i,\max} + \mathbb{E}\left[\, t_{l_{\max}} + l_{\max} + N^2 \mid \neg\mathcal{F}\right] \cdot \Delta_{i,\max} + 2NK\Delta_{i,\max} \tag{7}$$

$$\leq N\Delta_{i,\max} + \left(T^{\frac{1}{2-2^{1-q(S,K)}}}\sqrt{\frac{96K\,\log T}{\Delta^2}} + 1 - \log_2\left(1 - \frac{1}{2\log T^{\frac{1}{2-2^{1-q(S,K)}}}}\ln\left(\frac{96K\log T}{\Delta^2}\right)\right)\right) \cdot \Delta_{i,\max}$$

$$+ N^2\Delta_{i,\max} + 2NK\Delta_{i,\max}, \tag{8}$$

similar to Appendix C.1, where Eq.(6) comes from Lemma C.1, Eq.(7) holds according to Algorithm 3 and Lemma C.2, Eq.(8) holds based on Lemma C.7.

**Lemma C.7.** *Conditional on $\neg\mathcal{F}$, phase 2 will proceed in at most $l_{max}$ batches where*

$$l_{max} = \min\left\{l \; : \; t_l \geq 96K\log T/\Delta^2\right\},$$

*which implies that $t_{l_{max}} \leq T^{\frac{1}{2-2^{1-q(S,K)}}}\sqrt{96K\log T/\Delta^2}$ and $l_{max} \leq 1 - \log_2\left(1 - \frac{1}{2\ln(T^{\frac{1}{2-2^{1-q(S,K)}}})}\ln\left(\frac{96K\log T}{\Delta^2}\right)\right)$. And all players will enter phase 3 simultaneously at time grid $l_{max}$.*

*Proof.* Since

$$t_{l_{\max}} \geq 96K\log T/\Delta^2,$$

We have

$$t_{l_{\max}-1} \leq 96K\log T/\Delta^2.$$

According to the recursion formula $t_l = a\sqrt{t_{l-1}}$ we get

$$t_{l_{\max}} = T^{\frac{1}{2-2^{1-q(S,K)}}}\sqrt{t_{l_{\max}-1}} \leq T^{\frac{1}{2-2^{1-q(S,K)}}}\sqrt{96K\log T/\Delta^2}.$$

Substitute $t_l = T^{\frac{2-2^{1-l}}{2-2^{1-q(S,K)}}}$ into the above inequality we get

$$T^{\frac{2-2^{1-l_{\max}}}{2-2^{1-q(S,K)}}} \leq T^{\frac{1}{2-2^{1-q(S,K)}}}\sqrt{96K\log T/\Delta^2}.$$

Solving the inequality for $l_{\max}$ we get

$$l_{\max} \leq 1 - \log_2\left(1 - \frac{1}{2\ln(T^{\frac{1}{2-2^{1-q(S,K)}}})}\ln\left(\frac{96K\log T}{\Delta^2}\right)\right).$$

$\square$

## C.3 PROOF OF THEOREM 4.1(C)

Denote $\mathcal{F}_d^{(t)}$ as the event that $\mathrm{LCB}_{i,\sigma_{i,k}}(t) > \mathrm{UCB}_{i,\sigma_{i,k+1}}(t)$ for all $k \in [N]$, and $\mathrm{LCB}_{i,\sigma_{i,N}}(t) > \mathrm{UCB}_{i,\sigma_{i,k}}(t)$ for all $N+1 \le k \le K$. We have

$$Reg_i^\alpha(T) = \mathbb{E}\left[\alpha T \cdot \mu_{i,m_i^*} - \sum_{i=1}^T X_i(t) \mid \mathcal{F}\right] \cdot \mathbb{P}(\mathcal{F})$$

$$+ \mathbb{E}\left[\alpha T \cdot \mu_{i,m_i^*} - \sum_{i=1}^T X_i(t) \mid \neg\mathcal{F}\right] \cdot \mathbb{P}(\neg\mathcal{F})$$

$$\le \alpha T \cdot \mathbb{P}(\mathcal{F}) + \mathbb{E}\left[\alpha T \cdot \mu_{i,m_i^*} - \sum_{i=1}^T X_i(t) \mid \neg\mathcal{F}\right] \tag{9}$$

$$\le 2\alpha NK + \mathbb{E}\left[\alpha T \cdot \mu_{i,m_i^*} - \sum_{i=1}^T X_i(t) \mid \neg\mathcal{F}\right] \tag{10}$$

$$\le 2\alpha NK + \mathbb{E}\left[\left(\alpha T \cdot \mu_{i,m_i^*} - \sum_{i=1}^T X_i(t)\right) \mathbb{1}\{\mathcal{F}_d^{(T_0)}\} \mid \neg\mathcal{F}\right]$$

$$+ \mathbb{E}\left[\left(\alpha T \cdot \mu_{i,m_i^*} - \sum_{i=1}^T X_i(t)\right) \mathbb{1}\{\neg\mathcal{F}_d^{(T_0)}\} \mid \neg\mathcal{F}\right]$$

$$\le 2\alpha NK + \alpha T_0 + \mathbb{E}\left[\left(\alpha T \cdot \mu_{i,m_i^*} - \sum_{i=1}^T X_i(t)\right) \mathbb{1}\{\neg\mathcal{F}_d^{(T_0)}\} \mid \neg\mathcal{F}\right], \tag{11}$$

where Eq.(9) comes from $\mu_{i,m_i^*} \le 1$ and $X_i(t) \ge 0$. Eq.(10) holds according to Lemma C.1. Eq.(11) comes from the fact that if the good event $\neg\mathcal{F}$ happens and the player has an accurate estimation of his preference list before $T_0$, i.e., $\mathcal{F}_d^{(T_0)}$ happens, the Gale-Shapley algorithm would return a player-optimal stable matching. Since $\alpha \in (0, 1]$, the approximation regret would be less than $\alpha T_0 + (\alpha - 1) \cdot (T - T_0) \le \alpha T_0$.

When $\neg\mathcal{F}_d^{(T_0)}$ happens, we have overlaps in the confidence intervals from some player $p_i$'s estimated preference list, then we would call the offline approximation oracle $\mathbb{O}$ according to Algorithm 3. Moreover, conditional on $\neg\mathcal{F}$, we have that, for any $i \in [N]$ and $j \in [K]$, $|\hat{\mu}_{i,j}(t) - \mu_{i,j}| \le \sqrt{\frac{6\log T}{T_{i,j(t)}}}$. According to the detailed discussion of the properties of the approximation oracle $\mathbb{O}$ in Lin et al. (2024), we have that

$$\alpha\mu_{i,m_i^*} - \mathbb{E}X_i(t) \le 2\sqrt{\frac{6K\log T}{T_0}}. \tag{12}$$

Since there are $T - T_0$ rounds to pull, we have

$$\mathbb{E}\left[\left(\alpha T \cdot \mu_{i,m_i^*} - \sum_{i=1}^T X_i(t)\right) \mathbb{1}\{\neg\mathcal{F}_d^{(T_0)}\} \mid \neg\mathcal{F}\right] \le \alpha T_0 + 2\sqrt{\frac{6K\log T}{T_0}}(T - T_0). \tag{13}$$

Therefore, combining Eq.(12) and (13), we have

$$Reg_i^\alpha(T) \le 2\alpha NK + 2\alpha T_0 + 2\sqrt{\frac{6K\log T}{T_0}}(T - T_0). \tag{14}$$

Substitute $T_0 = T^{2/3}(K\log T)^{1/3}$ into Eq.(14), we have

$$Reg_i^\alpha(T) \le O((K\log T)^{\frac{1}{3}}T^{\frac{2}{3}}).$$

## C.4 PROOF OF REMARK 4.1

The switching cost bounds in Theorem 4.1(a) and Theorem 4.1(b) reflect worst-case scenarios where the exploration part persists for the maximum possible number of phases, namely $q(S, K)$. However, in expectation, the algorithm achieves substantially lower switching costs, as shown below.

We first consider the geometric instantiation. By Lemma C.4, all players enter part 3 simultaneously after at most $l_{\max}$ exploration phases, where

$$l_{\max} \leq \frac{1}{\log T^{\frac{1}{q(S,K)}}} \log \left( T^{\frac{1}{q(S,K)}} \cdot \frac{96K\log T}{\Delta^2} \right).$$

This bound implies that the exploration halts earlier than the worst-case schedule with high probability. Consequently, the cumulative number of switches grows only logarithmically in the effective horizon, yielding an expected switching cost of order

$$O\left( \log \frac{\log T}{\Delta^2} \right).$$

Turning to the minimax instantiation, Lemma C.7 establishes that part 2 consists of at most $l_{\max}$ phases, with

$$t_{l_{\max}} \leq T^{\frac{1}{2-2^{1-q(S,K)}}} \sqrt{96K\log T/\Delta^2}.$$

This sharper control of the phase length ensures that the number of switches grows doubly-logarithmically in the effective horizon. Hence, the expected switching cost improves to

$$O\left( \log \log \frac{\log T}{\Delta^2} \right).$$

In summary, while Theorem 4.1 provides conservative worst-case guarantees, Lemma C.4 and Lemma C.7 together imply that the expected switching cost is significantly smaller: logarithmic for the geometric instantiation, and doubly-logarithmic for the minimax instantiation. These results justify the exception costs stated in Remark 4.1.

### C.5 PROOF OF THEOREM 5.1

Let $\tau^{\mathrm{ACH}}$ denote the Hamiltonian circuit returned by the $\beta$-approximate TSP solver on the aggregated graph $\bar{G}$. By construction, each player $i$ follows a rotation of $\tau^{\mathrm{ACH}}$, so

$$\mathrm{ALG} = \bar{C}(\tau^{\mathrm{ACH}}) \leq \sum_{i=1}^{N} C_i(\tau^{\mathrm{ACH}}).$$

Since the solver is a $\beta$-approximation, we have

$$\sum_{i=1}^{N} C_i(\tau^{\mathrm{ACH}}) \leq \beta \cdot \mathrm{OPT}_{\bar{G}}, \tag{15}$$

where $\mathrm{OPT}_{\bar{G}}$ is the minimum aggregated TSP cost on $\bar{G}$.

Now let $\mathcal{T}' = \{\tau_1', \ldots, \tau_N'\}$ be the optimal solution for each player $p_i$'s TSP instance and denote $\mathrm{OPT}_i$ as the cost of each player's optimal solution. Let

$$j = \arg\min_i \mathrm{OPT}_i, \tag{16}$$

where $j$ represents the index of the smallest optimal solution cost. Therefore we have

$$\mathrm{ALG} \leq \beta \cdot \bar{C}(\tau_j') \tag{17}$$

$$= \beta \cdot \sum_{i=1}^{N} C_i(\tau_j')$$

$$\leq \beta \cdot \sum_{i=1}^{N} \gamma \cdot C_j(\tau_j') \tag{18}$$

$$\leq \beta \cdot \gamma \sum_{i=1}^{N} \mathrm{OPT}_i \tag{19}$$

$$\leq \beta \cdot \gamma \cdot \text{OPT}_{\text{par}}, \tag{20}$$

where Eq.(17) comes from Eq.(15), Eq.(18) is derived from Assumption 5.1 and Eq.(19) comes from Eq.(16). Eq.(20) comes from a trivial relationship that the sum of each player's optimal TSP instance cost is no more than the optimal total cost of the Parallel TSP.

## D    TECHNICAL LEMMAS

**Lemma D.1.** *(Corollary 5.5 in Lattimore & Szepesvári (2020)) Assume that $X_1, X_2, \ldots, X_n$ are independent, $\sigma$-subgaussian random variables centered around $\mu$. Then for any $\epsilon > 0$,*

$$\mathbb{P}\left(\frac{1}{n}\sum_{i=1}^{n} X_i \geq \mu + \epsilon\right) \leq \exp\left(-\frac{n\epsilon^2}{2\sigma^2}\right), \quad \mathbb{P}\left(\frac{1}{n}\sum_{i=1}^{n} X_i \leq \mu - \epsilon\right) \leq \exp\left(-\frac{n\epsilon^2}{2\sigma^2}\right).$$

## E    THE USE OF LLMS

We utilized a Large Language Model (LLM) as a writing assistant in the preparation of this manuscript. The LLM's role was strictly limited to improving the clarity, grammar, and readability of the text. All core scientific contributions—including the initial ideation, problem formulation, theorem development, mathematical proofs, algorithm design—were exclusively conceived and executed by the authors. The LLM was not used for generating any substantive content or research ideas.

