# OpenReview forum: "Bandit Learning in Matching Markets with Switching Cost"
_ICLR.cc/2026/Conference — ICLR 2026 Conference Withdrawn Submission_

### Official Review · Reviewer_CTwE · 2025-10-25

**Soundness:** 2
**Presentation:** 3
**Contribution:** 2
**Rating:** 2
**Confidence:** 3

**Summary:**

This work studies bandit learning in two-sided matching markets. Prior work achieves sublinear regret but incurs $O(T)$ switching costs over a horizon $T$. Such frequent reassignments are impractical in real-world applications due to operational and coordination costs. To address this issue, the paper aims to minimize regret while keeping switching costs low.

Under unit switching costs, the authors claim to match existing regret bounds while reducing the number of switches to $O(\log T)$. Moreover, they show that by further reducing switching to $O(\log \log T)$, the regret becomes $O(\sqrt{T}/\Delta^{2})$. In the extreme case of only $O(1)$ switches, they obtain a $T^{2/3}$ ($\alpha$-fraction) regret bound. The paper also discusses a generalization of their approach to heterogeneous switching costs.

**Strengths:**

This work provides the first algorithm that achieves low switching costs for bandit learning in two-sided matching markets.

**Weaknesses:**

1. A primary concern is the tightness of the regret bounds. In the literature on batched bandits, it is well known that near-optimal regret can be achieved with only $O(\log \log T)$ switching cost. However, in this work, the proposed algorithm attains a suboptimal regret of $\tilde{O}(\sqrt{T}/\Delta^{2})$ under the same switching budget.

2. Another concern relates to the novelty of the algorithmic contribution. Although it is interesting to explore low-switching algorithms in two-sided matching bandits, it is unclear what new ideas are introduced beyond existing epoch-based batched bandit techniques and prior work on matching-market bandits.

3. Finally, in the heterogeneous switching cost setting, although the paper claims to obtain theoretical guarantees analogous to the unit-cost case, the regret bounds are not clearly specified. A more explicit statement of the regret order in this setting would significantly improve clarity.

**Questions:**

1. Could the authors better clarify the novelty of their algorithmic contributions? In particular, it is not fully clear what elements go beyond standard epoch-based batched bandit techniques and prior matching-bandit algorithms.

2. Could you elaborate more on the regret guaranteed under heterogeneous switching costs?


3. In Algorithm 1, it is unclear how the total number of rounds is controlled. Specifically, Line 8 runs $(t_{l+1} - t_l)/K$ consecutive pulls per arm in each phase, which appears to exceed the total horizon $T$ when summing across all phases. A clarification of how the time budget aligns with the horizon would be helpful.

---

> ### Author Response · Authors · 2025-12-03
>
> We thank the reviewer for recognizing the potential of our problem formulation and framework.
>
> - On the Novelty of the Method
>
> We would like to clarify that our method is not a direct adaptation of existing works but rather a principled framework that emerges as a convergent solution to address the core challenges of matching markets with switching costs.
>
> We face a new three-way trade-off in this problem: learning efficiency, conflict among multiple players, and the switching cost. Our design process was driven by the need to resolve this fundamental trade-off:
>
>   (i) The dilemma of adaptive exploration: For classic bandit algorithms such as the upper confidence bound (UCB) and Thompson sampling (TS), their arm-switching patterns are inherently data-dependent and thus cannot yield a priori bounds on switching counts. Also as analyzed in matching markets (Liu et al. 2020, Kong et al., 2022), such algorithms cannot prevent players' frequent conflicts and only converge to the pessimal stable matching.
> To ensure a predictable and controllable switching cost, the most direct and effective approach is to explore a single arm consecutively for a period of time. And to avoid frequent conflicts among players, each player needs to simultaneously keep $K$ candidate arms in rotation.
>
>   (ii) The balance between learning efficiency and switching cost: Regarding switching cost, the most ideal case is switching exactly K times before finding the stable matching. But this approach severely hurts learning efficiency, as it wastes too much time on exploration before making exploitation decisions.To determine when exploration is sufficient, a periodic evaluation mechanism is required. Therefore, structuring the exploration into batches and evaluating after each one becomes the most natural and logical choice.
>
> When these two design principles—consecutive exploration and batched monitoring—are combined, the resulting framework naturally takes the form of a round-robin scheme. This is not a matter of simple imitation, but rather a necessary outcome of resolving the fundamental trade-off between learning efficiency, player conflict, and switching cost. Within this general framework, selecting the phase length is far from straightforward: different choices lead to distinct trade-offs between switching cost and regret. To address this, we propose several principled strategies for determining phase length. Although the geometric and single-batch cases may superficially resemble prior work, this resemblance is a convergent consequence of optimizing stable regret and optimizing switching cost for our new setting. **We believe that algorithmic value should not be judged solely by superficial novelty in appearance: achieving both provably reduced switching cost and regret guarantees in a challenging new multi-player setting is a meaningful and substantive contribution in itself.**
>
> - On the regret guarantees under heterogeneous switching costs
>
> For clarity, the paper first analyzes the unit-cost case, where each switch costs 1 and the budget $S$ determines the number of exploration phases $q(S,K)$. Theorem 4.1 shows that under geometric and minimax grids we obtain
> \\[
> Reg_i(T)
> = O\\Big(T^{1/q(S,K)} \tfrac{K\log T}{\Delta^2}\Big)
> \quad\text{and}\quad
> Reg_i(T)
> = O\\Big(T^{1/(2-2^{1-q(S,K)})}\sqrt{\tfrac{KT\log T}{\Delta^2}}\Big),
> \\]
> so when $q(S,K)$ is large enough (e.g. $S = \Omega(K\log T)$) these bounds reduce to the usual
> $\tilde O(\tfrac{K\log T}{\Delta^2})$ and $\tilde O(\sqrt{KT/\Delta^2})$ rates.
>
> In the **heterogeneous** switching-cost setting, $ArmSequencer()$ produces collision-free tours with total cost $ALG$, which is within a constant factor of the optimal parallel TSP cost $OPT_{\mathrm{par}}$. Given a budget $S$, the effective number of exploration phases becomes
> \\[
> M = \Big\lfloor \frac{S - \max_{i,j,j'} c_{i,j,j'}}{ALG} \Big\rfloor,
> \\]
> and the regret bounds above carry over with $q(S,K)$ replaced by $M$, up to constant factors depending on the approximation ratio and the heterogeneity parameter. Intuitively, the heterogeneous-cost case therefore inherits the same regret--switching trade-off as the unit-cost case; a larger budget $S$ (relative to $ALG$ or $OPT_{\mathrm{par}}$) increases $M$ and recovers the optimal logarithmic / $\sqrt{T}$ dependence on $T$.

---

> ### Author Response · Authors · 2025-12-03
>
> - On the total number of rounds in Algorithm 1
>
> We thank the reviewer for pointing out this potential source of confusion. In our algorithm, the time horizon is first split into phases by a predefined grid of time points (either geometric or minimax). Phase $l$ occupies the interval from $t_l$ to $t_{l+1}$, so its total length is $t_{l+1} − t_l$ rounds. In Line 8 we simply divide this phase length evenly across the $K$ arms, and each arm is pulled consecutively for $(t_{l+1} − t_l) / K$ rounds within this interval.
>
> We choose the grid so that the last endpoint $t_{M+1}$ is at most the horizon $T$, and we stop at the first phase index $l^\star$ with   $t_{l^*+1} \ge T$ (truncating the last phase if necessary). As a result, the total number of rounds used by all phases together is at most $T$, and Algorithm 1 never exceeds the horizon. We will clarify this stopping rule in the revised version and slightly adjust the pseudo-code indentation to emphasize that the inner loop only fills a fixed time interval $[t_l, t_{l+1})$ rather than adding extra rounds on top of $T$.

---

### Official Review · Reviewer_KWvM · 2025-10-29

**Soundness:** 3
**Presentation:** 3
**Contribution:** 2
**Rating:** 4
**Confidence:** 3

**Summary:**

The paper investigates the bandit learning problem in a two-sided matching market where one side (the players) has unknown preferences (utilities) regarding the other side (the arms). The optimal stable matching must be learnt over a time horizon T. Crucially, the authors move beyond the standard assumption of cost-free switching, which allows for potentially O(T) reassignments. By explicitly modeling switching costs, the paper aims to minimize the player-optimal stable regret under a predefined switching cost budget. To address this problem, the authors propose new algorithms (such as SCAM) and provide corresponding theoretical regret analyses.

**Strengths:**

**1. Theoretical Rigor and Coherence:** The paper is well-written and logically sound. The arguments are self-contained, and the proofs appear complete. I have read through the proofs and did not find any fundamental flaws or fatal errors in the analysis.

**2. Meaningful Motivation:** I endorse the paper's motivation. The concept of switching cost in matching markets is essential; in real-world scenarios (e.g., job assignments, resource allocation), switching a match involves various extra overheads, disruptions, and administrative costs. Incorporating this constraint makes the model more practical than existing works.

**Weaknesses:**

**1. Lack of Experimental Validation:** The paper is purely theoretical, and the absence of any experimental evaluation on real-world datasets is concerning. Without experiments to demonstrate the performance of the proposed algorithm (especially concerning the trade-off between regret minimization and cost budget adherence) against baselines in real-world scenarios, the structure of the article feels incomplete and unpolished. The lack of experiments also raises a strong suspicion regarding the real-world applicability of this model. It leaves the impression that the scenario modeled in this paper might be purely hypothetical and not grounded in a verifiable practical context.

**2. Insufficient Technical Novelty:** The two-sided market bandit model, the incorporation of switching costs, and the analytical techniques employed are all fairly conventional and do not offer substantial novelty for a paper submitted to ICLR 2026. The core algorithmic ideas—namely, grouping exploration rounds for the same arm and adopting exponentially increasing epoch lengths (the doubling trick) to reduce switching—are standard practices in the broader bandit literature when addressing switching cost constraints. While the paper is clearly written and the analysis is sound, I did not find the technical contributions particularly innovative. Overall, the work appears to be a rather straightforward application of existing methodologies developed for matching market bandit problems.


**3. Minor**  There appears to be a structural issue in the description of Algorithm 1. Specifically, the loop in Line 7 seems to be incorrectly nested within the loop in Line 5. This is, however, a minor issue and should be easy to fix.

**Questions:**

1. Could you please give an example and explain in detail what real-world data set you would choose if you were to do experimental verification, how you would process the data, and how you would design the experiment?
2. Have you considered how to derive results concerning the regret lower bound, and how to construct a hard instance to prove it?

---

> ### Author Response · Authors · 2025-12-03
> **On potential real-world experimental evaluation**
>
> We appreciate the suggestion to discuss a concrete empirical setup. Our current submission is theory-focused, but we agree that an experimental study would be valuable and outline here how we would design it.
>
> A natural real-world source of data is a two-sided platform such as student–project or course allocation, where students repeatedly interact with projects/courses and both sides have preferences. Public datasets of this type exist (e.g., student–project allocation data with real student preferences and project capacities). Another option is online labor markets such as Upwork, for which large-scale job–freelancer datasets are available.
>
> In either case, we would proceed as follows.
>
> • Data preprocessing. We select a subset of users on each side with sufficient history. For each user–item pair (student–project, worker–job) we estimate an empirical mean reward (e.g., normalized grade, completion quality, or rating) and use this as $μ_{i,j}$; priorities on the item side (projects, jobs) are inferred from historical admission or hiring patterns.
>
> • Switching-cost model. We define a heterogeneous switching cost $c_{i,j,j′}$ from item $j$ to $j′$ using observable features, such as change in topic area, supervisor, schedule, or job type. This yields a realistic cost matrix and allows us to set budgets S that correspond to a small number of reassignments per user over the horizon.
>
> • Experimental protocol. Using the inferred $μ_{i,j}$, priorities, and $c_{i,j,j′}$, we simulate repeated matching over $T$ rounds. At each round the algorithm produces a matching, we sample stochastic rewards from the empirical distributions, and we track both (player-optimal) stable regret and cumulative switching cost. We would compare our SCAM variants (geometric/minimax/single-phase, ACH/TPE) to existing matching-bandit baselines that ignore switching costs or use simple block-ETC style exploration, under varying budgets $S$.

---

> ### Author Response · Authors · 2025-12-03
>
> - Tightness of the result:
>
> To assess the tightness of our result, we draw on insights from the single-player setting—specifically, the classical multi-armed bandit (MAB) model. Our goal is to leverage well-established results in the MAB literature to validate our theoretical guarantees. Prior work [1] has shown that achieving the optimal regret of $O(K \log T / \Delta)$ in the MAB setting requires at least $\Omega(\log T)$ batches. Our result recovers this case when reduced to the single-player setting and matches the known lower bound, thereby supporting the tightness of our analysis.
> We agree that providing a lower bound for more general matching markets can better illustrate the tightness and we leave this as the next plan of our work.
>
> [1] Perchet V, Rigollet P, Chassang S, Snowberg E (2016). Batched Multi-armed Bandits Problem. The Annals of Statistics 44(2):660681.

---

### Official Review · Reviewer_H1hi · 2025-11-03

**Soundness:** 3
**Presentation:** 3
**Contribution:** 2
**Rating:** 4
**Confidence:** 3

**Summary:**

This paper considers matching in bandit markets with switching costs.
The algorithm uses a phased design, similarly to successive elimination in bandit problems. It also consecutive plays each arm before moving on to the next arm. I wonder if this rather obvious adaptation is taken into account when considering existing algorithms. Let's take the ETC algorithm from Lui et al. It performs $hK$ matchings, and so an equal number of switches. If, instead of cyclic mathings, it performed them in blocks, then it would have only $h$ switches. Setting $h = O(\ln(N T))$. So,  $N,K$ do not appear outside the logarithm, which makes sense.

In general, I find these 'decentralised' extensions of the matching problem to be pseudo-decentralised. They rely on an initial agreement to follow a protocol which has to be centrally managed in order for arms and players to obtain indices. In this setting, I do not see why players could simply not re-propose until they get matched, for example: with the matchings taking a very short amount of time, relative to the time needed to obtain the reward. In the end, though, why do we even get rid of the matching platform? The whole point of the matching problem is to make sure it matches everybody according to their preferences.

The basic algorithm is simple enough. After indexing to avoid conflicts, there are phases within which each player can sample up to K arms. To simplify, at subphase l, player i is playing arm s(i,l) so that none are playing the same arm. This setting is a rather straightforward extension.

The general cost case requires solving a parallel TSP problem.
If we are talking about sample complexity, switching and regret, then I think we do not really care about computation: I think the discussion about optimal PTSP solutions is beside the point: the authors could have simply said they assume they obtain an approximate PTSP solution somehow. Again, this problem would not have arisen in the centralised case, and it somehow seems to me as though they authors are trying to make the problem more convoluted for no good reason.

**Strengths:**

+ Very well written
+ OK overview of prior work, including switching.
+ The general switching cost setting has some more complex elements.

**Weaknesses:**

- The prior work ignored the switching problem, but it was trivial to fix in most cases. In some sense, this misrepresents the prior work.
- The extension then seems almost trivial in the uniform cost setting.
- While general cost setting is valid, it becomes contrived once you factor in decentralised proposals.

**Questions:**

Why do we need to be 'decentralised'?
Why did you not take into account the trivial method for taking switching into account in prior work?
Why do you complexify the general cost case and not simply assume you have an epsilon-optimal solution?

---

> ### Author Response · Authors · 2025-12-03
>
> ### (1) **Clarification on the ETC baseline of Liu et al. (2020).**
> The reviewer suggests that one could take the centralized Explore–Then–Commit (ETC) algorithm of Liu et al. (2020), group the cyclic explorations into blocks, and choose $h=O(\log(NT))$ to obtain a small number of switches. We would like to clarify why this adaptation is not immediate and does not address our setting.
>
> In Liu et al.’s work, the centralized ETC planner takes as inputs an exploration budget $h$ and the horizon $n$ and performs $hK$ exploration matchings before committing. Their regret bound(Theorem 1) is obtained by choosing$$ h =\max \left\\{    1,\frac{4}{\Delta^2}\log\left(1+\frac{n\Delta^2N}{4}\right)  \right\\},$$ where $\Delta$ is the global minimum reward gap across all players and arms. They explicitly note that, as in the single-agent case, centralized ETC requires knowledge of both the horizon $n$ and the minimum gap $\Delta$. Moreover, they point out that known adaptive stopping rules for single-agent ETC cannot be implemented in their matching-market setting because the platform does not observe agents’ rewards.
>
> In contrast, our batched–style algorithms are designed for a different constraint: we are given a switching-cost budget $S$, and the algorithm automatically determines the number and lengths of epochs based on $S$, without knowing the horizon $T$ or any reward gaps in advance. As in standard bandit analysis, the regret bounds in Theorem 4.1 are stated in terms of $t$ and the gaps, but these quantities are used only in the analysis: for any fixed unknown $T$ the algorithm can be run with a given budget $S$, and the theorem guarantees a certain regret-switching trade-off, without tuning any parameter during execution.
>
> ### (2) **Decentralization vs. central coordination**
> Thank you for raising this point. In our paper, “decentralised’’ does **not** mean that there is literally zero communication; rather, we follow the standard usage in the decentralized matching-bandit literature: players have **no direct communication channel** and base their learning and decisions only on their own sequence of matches, collisions and rewards [1].
>
> In our ETGS-style algorithms, coordination is limited to **batch boundaries**: a common clock / platform signal announces the end of a phase and (in the centralized GS variant) runs a single Gale–Shapley call using the players’ current indices. Between these synchronization points, each player selects arms and updates confidence bounds purely from its own observations; players never see others’ rewards, and there is no per-round centralized recomputation of matchings. Overall, apart from one offline preprocessing step (ArmSequencer / ACH) and $M$ such synchronization signals, learning and arm selection are fully local, which is why we refer to the online part of the algorithm as decentralized.
>
> This modelling choice is motivated by large matching platforms (online labor markets such as Upwork/TaskRabbit, ride-sharing, etc.). In these  systems, individual users on both sides hardly  coordinate or share detailed reward information; they most likely observe only their own sequence of matches, collisions and payoffs. The decentralized matching-bandit literature explicitly adopts this view: for example, Liu et al. [1] assume that players “have no direct means of communication’’ in decentralized matching markets, and show that this more realistically captures large platforms than a central-clearinghouse model. Subsequent works on fully-decentralised two-sided matching bandits  also emphasize that online learning must be done locally, without a central organizer, precisely because real platforms do not coordinate agents at every step.
> At the same time, many platforms **can** perform limited centralized planning _before_ interaction begins (e.g., computing a good exploration schedule using prior information). Our framework is designed for exactly this regime: the platform uses ex-ante knowledge of switching costs to compute the tours once, and then the subsequent learning and matching are executed in a decentralized manner by the players. We will make this “one-time centralized preprocessing + decentralized online learning’’ structure explicit in the revised paper.
>
> [1] Lydia T. Liu, Feng Ruan, Horia Mania, Michael I. Jordan. Bandit Learning in Decentralized Matching Markets. JMLR 22(211):1−34, 2021.

---

> > ### Author Response · Authors · 2025-12-03
> >
> > ### (3) **Why choose the switching cost as a hard constraint**
> >
> > We acknowledge that some prior works incorporate the switching cost directly into the per-round regret, treating it on equal footing with the received reward.
> >
> > In contrast, our approach deliberately separates the switching cost from the reward. This modeling choice is motivated by practical scenarios where switching is subject to hard constraints—such as physical limitations or budgetary caps that restrict the number of allowable switches. In such settings, embedding switching costs directly into the regret lacks advantage, as it makes it difficult to disentangle the impact of switching from that of reward accumulation.
> >
> > Furthermore, our theoretical results explicitly characterizes the trade-off between regret and switching frequency. This formulation offers actionable insights for real-world applications where balancing performance and operational cost is critical. By making this trade-off transparent, our framework empowers practitioners to adapt algorithms to domain-specific requirements and constraints.
> >
> > ### (4)**Why we formulate PTSP and solve it**
> >
> > Given the decentralized matching–bandit setting discussed above, the platform’s only global control is to design an exploration schedule _before_ online interactions start. In the heterogeneous switching–cost case, this naturally leads to a **routing problem over players and arms**: we need a set of tours such that each player visits all arms, players never collide, and the _total_ switching cost is as small as possible under the budget. This is exactly a parallel variant of the probabilistic/multiple Travelling Salesman Problem (PTSP / mTSP), both of which are well-studied in operations research and routing (e.g., PTSP and its variants, multiple-TSP surveys and applications).
> >
> > Modelling the general-cost case explicitly as this PTSP-type problem has two advantages compared to simply postulating an ε-optimal oracle. First, our regret bounds depend _quantitatively_ on the tour cost (through the parameter ALG\text{ALG}ALG and its relation to the optimum), rather than hiding all offline complexity inside an unspecified ε. Second, the resulting “parallel PTSP with collision constraints’’ is itself a meaningful combinatorial optimisation problem beyond bandits, closely related to multi-agent routing and sustainable transportation. For these reasons, we choose to make the formulation explicit and provide a concrete heuristic with a provable approximation guarantee, while keeping the online analysis modular so that any alternative oracle with a given tour cost can be plugged into our framework.

---

### Official Review · Reviewer_HSSR · 2025-11-05

**Soundness:** 3
**Presentation:** 2
**Contribution:** 3
**Rating:** 6
**Confidence:** 3

**Summary:**

The paper considers the variant of the matching markets bandit problem where there is switching costs. The model is motivated by practical applications and other related works in online learning. The observation the paper makes is that most algorithms for matching markets work in two phases -- an exploration phase where every player learns their preferences over arms and an exploitation phase where players use the Gale Shapely matching based on their estimated arms. Existing algorithms incur a large switching cost in the exploration phase as they essentially employ round robin. The paper proposes an alternative for the exploration through carefully repeating each arm sufficient number of times. The exact number of times to play each arm in a phase is given by a geometric series. Through this analysis, the authors show that $O(K \log(T))$ switches can yield $O(K \log(t)/\Delta^2)$ regret. Further, the authors also generalize it to the case when switching costs are non-uniform. They show that a heuristic -- where they solve the problem for a single player and then offset the starting point for the other players, can yield similar results.

**Strengths:**

The paper has several theoretical contributions. It identifies a simple way to mitigate the round-robin style exploration common in all prior algorithms. Further, it shows a simple way of extending the analysis to the heterogeneous case, by utilizing connections to finding Hamilton paths in the cost graph. They solve the collision avoidance by a simple heuristic of computing the tour for a single player and then producing an offset for others. They have an approximation ratio showing that this is worse off compared to the optimal only by a fraction which is the ratio of the largest to the smallest switching-cost.

**Weaknesses:**

The paper's writing can be improved in two ways. First, it will make it better to read even with a few illustrative figures, especially for the Hamilton path heuristic and their core algorithmic contribution in the unit cost case. Second, it was not immediately obvious to me that if all costs are uniform, the generalized algorithm reduces to that of the unit-cost case. Having a section, maybe even in the Appendix can help readers parse and understand the algorithm better. Without these aids, the algorithms are quite hard to understand and parse.

**Questions:**

Can you explain/show, how the generalized algorithm reduces to the one you had for unit- switching cost case? This will help me understand the contributions of the generalized algorithm better.

---

> ### Author Response · Authors · 2025-12-03
> **Response to HSSR – connection between the generalized and unit-cost algorithms.**
>
> We thank the reviewer HSSR for the valuable comments and suggestions. Please find our detailed response below.
>
> In the paper we first present our Switching-Cost-Aware Matching (SCAM) algorithm in the **unit-cost** setting (Section 4) and then extend it to the **general heterogeneous-cost** setting (Section 5). Algorithmically, the online protocol is identical in both cases: SCAM has three components — (i) an estimation phase, (ii) a phased, consecutive exploration according to $ArmSequencer()$, and (iii) a final exploitation phase based on either Gale-Shapley or the approximation oracle.
>
> The only difference is how the exploration schedule is produced by $ArmSequencer()$, and the number of phases is computed from the switching-cost model and the budget.
> - In the unit-cost case, each switch has cost 1. A single "round-robin batch" of exploration for each player uses exactly $K-1$ switches, so under a per-player budget $S$ the number of phases is
> $$
> q(S,K)=\left\lfloor  \frac{S-1}{K-1}  \right\rfloor
> $$
> as stated in Section 4.
> - In the general-cost case, $ArmSequencer()$ first solves a Parallel Traveling Salesman Problem (PTSP) instance via the Aggregated Cost Heuristic (ACH, Algorithm 2), which returns a collision-free set of Hamiltonian circuits $\mathcal{T}^*$ with total cost ALG. Given a budget $S$ and the maximal single transition cost $\max_{i,j,j'} c_{i,j,j'}$, we set the number of exploration phases to $$
> M=\left\lfloor \frac{S-\max_{i,j,j'} c_{i,j,j'}}{ALG}\right \rfloor,
> $$
> ensuring that the cumulative switching cost doesn't exceed $S$.
>
> When all switching costs are uniform (each $c_{i,j,j'}=1$), the PTSP instance is symmetric, and ACH can return parallel round-robin tours as in the unit-cost case: every player visits all $K$ arms once per phase with $K-1$ switches. In this regime $ALG=(K-1)$ and $\max_{i,j,j'} c_{i,j,j'}=1$, so choosing a per-player budget $S$ and computing $M$ recovers the same effective number of exploration phases as $q(S,K)$ in Section 4, and the regret bounds in Theorem 4.1 reduces to the unit-cost guarantees.
>
> We will add a short remark and an explicit derivation in the appendix to make this reduction fully transparent.

---

### Note · Authors · 2026-01-18

**Comment:**

Dear Area Chairs and Reviewers,

We would like to thank the reviewers for their time and detailed feedback. After careful consideration, we have decided to withdraw this submission from ICLR.

The reviews highlighted several presentation and positioning issues, in particular the need for clearer exposition and illustrative figures for the offline coordination component, and a more explicit explanation of how the generalized (heterogeneous-cost) setting reduces to the unit switching-cost case when costs are uniform. We agree these points would materially improve the paper, and we plan to revise the manuscript accordingly, including a clearer reduction argument and more comprehensive empirical validation.

Thank you again for your constructive comments and suggestions.

Sincerely,
The Authors

**Withdrawal Confirmation:**

I have read and agree with the venue's withdrawal policy on behalf of myself and my co-authors.